# Chromatin binding by HORMAD proteins regulates meiotic recombination initiation

Carolyn R Milano[1,7], Sarah N Ur[2,6,7], Yajie Gu [2], Jessie Zhang [1], Rachal Allison [3], George Brown [3], Matthew J Neale [3], Eelco C Tromer [4], Kevin D Corbett [2,5✉] & Andreas Hochwagen [1✉]

## Abstract

**The meiotic chromosome axis coordinates chromosome organization and interhomolog recombination in meiotic prophase and is essential for fertility. In *S. cerevisiae*, the HORMAD protein Hop1 mediates the enrichment of axis proteins at nucleosome-rich islands through a central chromatin-binding region (CBR). Here, we use cryoelectron microscopy to show that the Hop1 CBR directly recognizes bent nucleosomal DNA through a composite interface in its PHD and winged helix–turn–helix domains. Targeted disruption of the Hop1 CBR-nucleosome interface causes a localized reduction of axis protein binding and meiotic DNA double-strand breaks (DSBs) in axis islands and leads to defects in chromosome synapsis. Synthetic effects with mutants of the Hop1 regulator Pch2 suggest that nucleosome binding delays a conformational switch in Hop1 from a DSB-promoting, Pch2-inaccessible state to a DSB-inactive, Pch2-accessible state to regulate the extent of meiotic DSB formation. Phylogenetic analyses of meiotic HORMADs reveal an ancient origin of the CBR, suggesting that the mechanisms we uncover are broadly conserved.**

**Keywords** Meiosis; HORMA; Chromosome Segregation; Budding Yeast; Chromatin
**Subject Categories** Cell Cycle; DNA Replication, Recombination & Repair; Structural Biology

## Introduction

Across eukaryotes, sexual reproduction requires meiosis, a two-stage cell division that produces haploid gametes from a diploid progenitor cell. The reduction of ploidy requires the recombination of homologous chromosomes to form interhomolog crossovers (COs), which facilitate chromosome alignment on the meiosis I spindle. COs result from the repair of programmed DNA double-strand breaks (DSBs) that are introduced throughout the genome at the onset of meiosis.

Failure to form sufficient DSBs or interhomolog COs can lead to errors in chromosome segregation and aneuploidy (Hassold and Hunt, 2001; Hassold et al, 2007).

Meiotic recombination occurs in the context of tightly choreographed changes in chromosome morphology that regulate the number and position of DSBs and CO repair events. These morphology changes are mediated in large part by proteins of the meiotic chromosome axis, or axial element. Axis proteins organize each replicated chromosome into a linear array of chromatin loops and help activate, distribute, and resolve meiotic recombination events (Ito and Shinohara, 2022; Ur and Corbett, 2021; Zickler and Kleckner, 2015). The axis comprises three main components: (1) cohesin complexes with one or more meiosis-specific subunits (Sakuno and Hiraoka, 2022); (2) filamentous axis core proteins (West et al, 2019); and (3) meiotic HORMA domain (HORMAD) proteins (Rosenberg and Corbett, 2015; Prince and Martinez-Perez, 2022). Cohesins are ring-like complexes that bind DNA and processively extrude chromatin loops to organize chromosomes throughout the cell cycle (Yatskevich et al, 2019). Axis core proteins bind cohesin complexes, mediating the assembly of extended coiled-coil filaments that anchor DNA loops (West et al, 2019). The filamentous core proteins bind HORMAD proteins which serve as master regulators of meiotic recombination and promote the stable alignment of recombining chromosomes by a zipper-like structure called the synaptonemal complex (SC) (Cahoon and Hawley, 2016; Grey and de Massy, 2022).

The function of HORMAD proteins in the activation and later repair of meiotic recombination is well conserved across eukaryotes but is best understood in the budding yeast *Saccharomyces cerevisiae* (Prince and Martinez-Perez, 2022). In *S. cerevisiae*, the HORMAD ortholog Hop1 contributes to axial element integrity (Klein et al, 1999; Hollingsworth et al, 1990) and acts in the very early stages of meiotic recombination by mediating the recruitment of the DSB-forming endonuclease Spo11 and its accessory proteins (Panizza et al, 2011; Woltering et al, 2000). Hop1 is then phosphorylated by the DNA damage-response kinases Tel1 and Mec1 (orthologs of mammalian ATM and ATR, respectively) to recruit the kinase Mek1 (Carballo et al, 2008), which biases DSB repair towards the homologous chromosome (rather than the sister chromatid) as a repair template (Niu et al, 2005, 2007). Once the synaptonemal complex assembles, the AAA+ ATPase Pch2 removes Hop1 from the axis (Börner et al, 2008; San-

[1]Department of Biology, New York University, New York, NY 10003, USA. [2]Department of Cellular and Molecular Medicine, University of California, San Diego, La Jolla, CA 92093, USA. [3]Genome Damage and Stability Centre, University of Sussex, Falmer BN1 9RQ, UK. [4]Groningen Biomolecular Sciences and Biotechnology Institute, University of Groningen, Groningen 9747 AG, The Netherlands. [5]Department of Molecular Biology, University of California, San Diego, La Jolla, CA 92093, USA. [6]Present address: Vividion Therapeutics, San Diego, CA 92121, USA. [7]These authors contributed equally: Carolyn R Milano, Sarah N Ur. ✉E-mail: kcorbett@ucsd.edu; andi@nyu.edu

Segundo and Roeder, 1999; Chen et al, 2014; Subramanian et al, 2016; Mu et al, 2020), thereby reducing DSB formation and promoting DSB repair to ultimately enable exit from meiotic prophase and entry into the first meiotic division.

The probability of DSB and CO formation on a given chromosome or chromosomal region varies dramatically across the *S. cerevisiae* genome, and this variation is tightly correlated with Hop1 distribution. Pericentromeric regions and regions flanking the ribosomal DNA, for example, both experience low levels of recombination and are both under-enriched for Hop1 (Sun et al, 2015; Chen et al, 2008; Mancera et al, 2008; Vader et al, 2011). In contrast, the shortest three chromosomes experience higher-than-average frequency of recombination and higher-than-average levels of Hop1 protein deposition (Chen et al, 2008; Mancera et al, 2008; Kaback, 1996). Notably, this chromosome size bias is dependent on Hop1, as recombination initiation factors on short chromosomes is abrogated in *hop1Δ* mutants (Murakami et al, 2021; Pan et al, 2011; Blitzblau et al, 2007; Sun et al, 2015; Murakami et al, 2020). Regional Hop1 enrichment, with a concurrent elevation of DSB formation and CO designation, is also observed in gene/nucleosome-dense blocks of chromatin distributed along the arms of all chromosomes termed "islands" (Heldrich et al, 2022).

The chromosomal binding and enrichment patterns of the axis core protein Red1 mirror those of its binding partner Hop1. In contrast, the Rec8-containing cohesin binding pattern is uniquely enriched in pericentromeric regions and is not enriched in islands. Hop1 drives the enrichment of Hop1/Red1 binding to islands, independently of Rec8. Accordingly, islands are the only genomic regions that retain Hop1/Red1 binding and DSB activity in *rec8Δ* mutants (Sun et al, 2015). Hop1 binding to islands requires a poorly characterized central region of Hop1 (Heldrich et al, 2022). This region harbors a previously identified zinc-binding domain (Hollingsworth and Byers, 1989), shown to be involved in Hop1 binding to double-stranded DNA in vitro (Hollingsworth and Byers, 1989; Kironmai et al, 1998), and is reminiscent of some PHD domains (Heldrich et al, 2022), which in other proteins directly bind nucleosomes (Sanchez and Zhou, 2011).

Here, we determined the structure and function of the Hop1 central region, which we termed the chromatin-binding region (CBR). The Hop1 CBR comprises tightly associated PHD and winged helix–turn–helix (wHTH) domains and directly recognizes the bent DNA on nucleosomes. Mutations that disrupt nucleosome binding cause loss of Hop1 enrichment and DSB activity from islands. Loss of CBR-nucleosome binding also makes cells highly dependent on Pch2, which we attribute to the dysregulation of distinct DSB-promoting and DSB-inactive conformations of Hop1. We identify highly divergent CBRs in meiotic HORMADs across diverse eukaryotes, suggesting that these functions may be broadly conserved, with likely functional specialization between eukaryotic lineages.

# Results

## Structure of the budding yeast Hop1 CBR

To understand the molecular basis for Hop1 CBR-mediated axis protein enrichment at axis genomic islands, we sought to determine a high-resolution structure of this domain. We could purify the isolated *S. cerevisiae* Hop1 CBR (residues 322–537) but could not

identify crystallization conditions. After screening the equivalent domain from several budding yeast Hop1 proteins, we determined a 1.5 Å-resolution crystal structure of the Hop1 CBR from *Vanderwaltozyma polyspora* (residues 317–535; 34% identical to *S. cerevisiae* Hop1 in this region) using single-wavelength anomalous diffraction methods (SAD) with the two natively bound zinc atoms ("Methods"; Appendix Table S1). The *V. polyspora* Hop1 CBR structure reveals a compact assembly with a PHD domain (residues 319–374) and a variant winged helix–turn–helix (wHTH) domain (residues 375–439) with a C-terminal extension that we term HTH-C (residues 440–524; Fig. 1A,B). The N-terminal PHD domain coordinates two zinc ions through seven conserved cysteine residues and one conserved histidine residue (Fig. 1B; Appendix Fig. S2) (Hollingsworth and Byers, 1989). The wHTH and HTH-C domains fold together, with the HTH-C region forming an elaborated β-sheet wing on the wHTH domain and a long C-terminal α-helix stretching across both the PHD and wHTH domains (Fig. 1B).

PHD domains are found in many families of chromatin-associated proteins, and typically bind modified lysine residues in histone tails via a conserved pocket lined with hydrophobic residues (Sanchez and Zhou, 2011). Examining our structure of the *V. polyspora* Hop1 CBR reveals that in Saccharomycetaceae Hop1 proteins, the PHD domain's canonical lysine binding pocket is poorly conserved and largely lacking hydrophobic residues (Appendix Fig. S1; Appendix Table S2). This lack of conservation casts doubt on whether the Hop1 PHD domain could bind a histone tail equivalently to other PHD domain proteins. Moreover, a comparison of the Hop1 CBR structure with those of DNA-bound wHTH domains reveals that in Hop1, the canonical wHTH DNA-binding surface is partially occluded by the PHD and HTH-C domains (not shown). Thus, if the Hop1 CBR directly binds nucleosome-rich chromatin as suggested by prior studies, it likely does so in a manner unlike canonical PHD or wHTH domain proteins.

## The Hop1 CBR binds nucleosomes

We next directly tested whether the Hop1 CBR binds to nucleosomes in vitro. Using a pulldown assay, we detected strong binding of the isolated *S. cerevisiae* Hop1 CBR to reconstituted mononucleosomes (Appendix Fig. S3A). To determine the structural basis for this interaction, we reconstituted a Hop1 CBR:mononucleosome complex (Appendix Fig. S3B) and collected a single-particle cryoelectron microscopy (cryo-EM) dataset (Appendix Fig. S4; Appendix Table S3). Initial 2D class averages clearly showed nucleosomes, with some classes showing weak density extending outward from the DNA gyres (Appendix Fig. S4). 3D reconstructions revealed two major particle classes corresponding to either a nucleosome or a Hop1 CBR:nucleosome complex. We refined the isolated nucleosome structure to a resolution of 2.57 Å, and the Hop1 CBR:nucleosome complex structure to 2.74 Å (Appendix Fig. S4; Appendix Table S3), enabling us to confidently build and refine complete atomic models and visualize the molecular details of the CBR-nucleosome interaction (Fig. 1C,D). Finally, we detected a minor population with two copies of the Hop1 CBR bound to neighboring locations on a single nucleosome. The low particle number of this class (25,000 particles, compared to 140,000 for the Hop1 CBR:nucleosome complex) precluded full

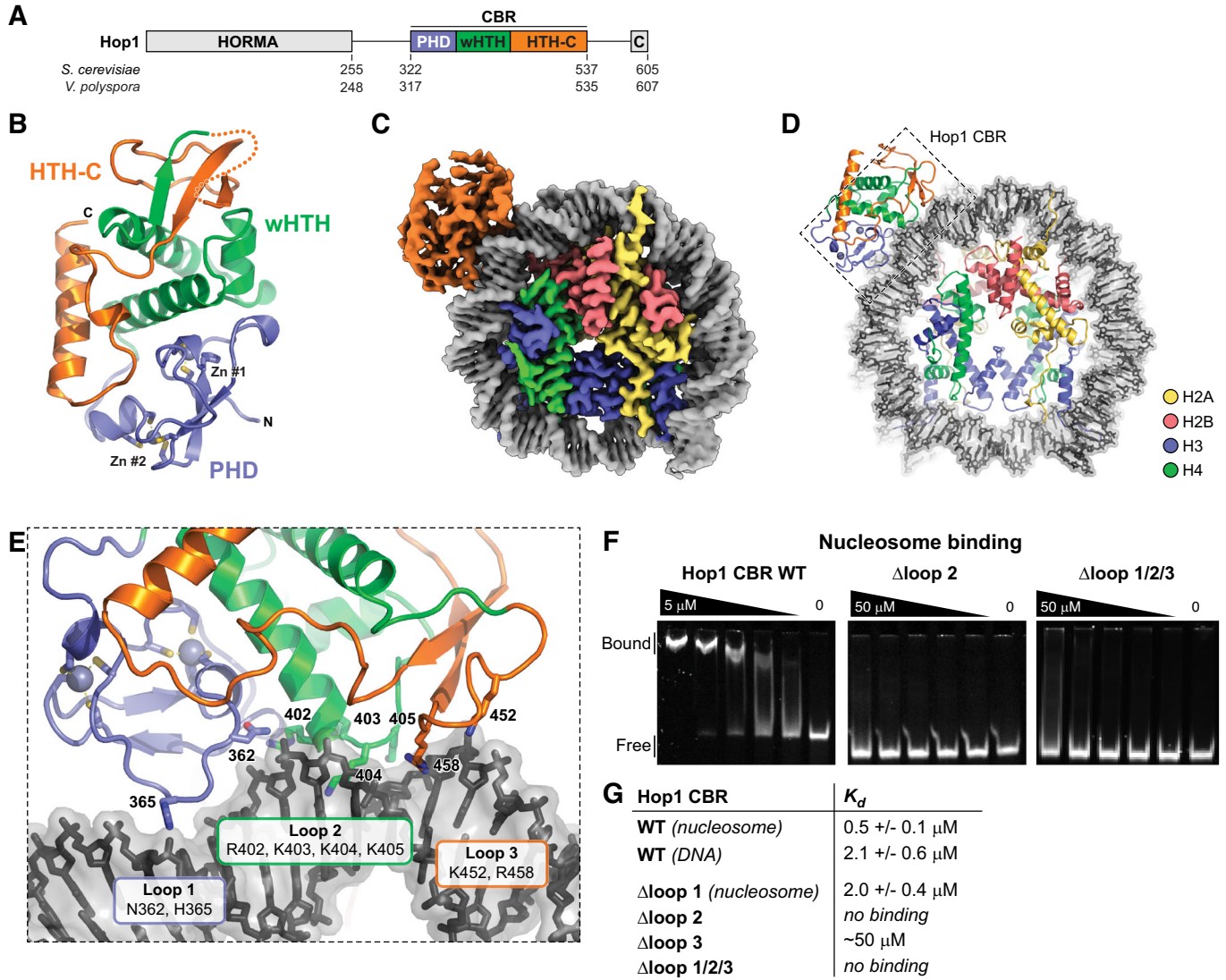

**Figure 1. Structure and nucleosome binding by the Hop1 CBR.**

(A) Domain structure of budding yeast Hop1. C closure motif. (B) Crystal structure of the *V. polyspora* Hop1 CBR, with PHD domain colored blue, wHTH green, and HTH-C orange. See Appendix Fig. S1 for sequence alignments, and Appendix Fig. S2 for sample electron density. (C) Cryo-EM density map at 2.74 Å resolution of *S. cerevisiae* Hop1 CBR bound to a human mononucleosome. (D) Cartoon view of *S. cerevisiae* Hop1 CBR bound to a mononucleosome, oriented as in (C). Hop1 is colored as in (B), DNA is colored gray/black, and histones are colored yellow (H2A)/red (H2B)/blue (H3)/green (H4). See Appendix Fig. S3 for reconstitution of the Hop1 CBR:nucleosome complex, Appendix Fig. S4 for cryo-EM structure determination workflow, Appendix Fig. S4 for structure of a nucleosome with two Hop1 CBRs bound, and Appendix Fig. S5 for comparison of *V. polyspora* and *S. cerevisiae* Hop1 CBR structures. (E) Closeup view of the Hop1 CBR-DNA interface, with key DNA-interacting residues of loops 1 (blue), 2 (green), and 3 (orange) shown as sticks and labeled. See Appendix Fig. S6 for cryo-EM density. (F) Representative electrophoretic mobility shift assays (EMSAs) showing binding of wild-type, Δloop2, and Δloop 1 + 2 + 3 mutants of *S. cerevisiae* Hop1 CBR to reconstituted human nucleosomes. The representative gel images are also shown in Fig. EV2. (G) Hop1 CBR-nucleosome and DNA-binding affinities determined from triplicate EMSA assays (see Fig. EV1 for gels). Source data are available online for this figure.

refinement and model building, but we could place two Hop1 CBRs into gaussian-smoothed maps at 3.15 Å resolution to understand the binding of both Hop1 CBR protomers to the nucleosome (Fig. EV1).

Nucleosomes contain two copies of each of four histone proteins —H2A, H2B, H3, and H4—with 146 bp of DNA wrapped ~1.75 times around the histone octamer core (Luger et al, 1997). While the histone octamer is two-fold symmetric, the "Widom 601" DNA sequence we used to reconstitute mononucleosomes is asymmetric

(Lowary and Widom, 1998). To determine the most likely orientation of this DNA sequence in our structure, we built and refined both possible orientations and chose the one with the higher overall Pearson's correlation coefficient between the final model and the experimental map. Within the nucleosome structure, locations along each 72–73 base stretch of DNA extending from the midpoint of the 146 bp DNA (the "dyad axis") are defined as superhelical locations (SHL) 0 at the dyad axis to +/− 7 near the DNA ends (Luger et al, 1997). In our structure, the major Hop1

CBR binding site is between SHL 2.5 and 3.5, on the outer edge of the nucleosome-wrapped DNA (Fig. 1E). The minor CBR binding site is located between SHL −5.5 and −6.5. The overall structure of the nucleosome-bound *S. cerevisiae* Hop1 CBR in our cryo-EM map closely matches our crystal structure of the *V. polyspora* Hop1 CBR (overall Cα r.m.s.d. of 1.06 Å; Appendix Fig. S5), with the most significant difference being that a disordered loop in the HTH-C domain of the *V. polyspora* Hop1 CBR is well-ordered and interacting with DNA in the nucleosome-bound *S. cerevisiae* Hop1 CBR (Fig. 1E).

The *S. cerevisiae* Hop1 CBR binds DNA through three loops, termed loops 1–3 (Fig. 1E; Appendix Fig. S7). Loop 1 is positioned in the PHD domain, with the side chains of N362 and H365 interacting with the major groove of base pairs 25–27 (as measured from the nucleosome dyad axis; all DNA base pairs noted are for the major Hop1 binding site spanning SHL 2.5–3.5). Loop 2 is in the wHTH domain, with R402, K403, K404, and K405 inserted into the minor groove of base pairs 28–31. Finally, loop 3 is in the HTH-C domain, with K452 and R458 binding base pairs 32–34. Overall, loops 1–3 define a ~10 bp footprint for the Hop1 CBR on the outer edge of the bent nucleosomal DNA. Notably, despite sharing a common fold with canonical DNA-binding wHTH domains, the Hop1 wHTH domain binds DNA on a distinct surface compared with these domains. Moreover, we do not detect any density for a histone tail bound to the Hop1 PHD domain. Together, these findings point to a model in which the Hop1 CBR recognizes nucleosomes solely through a non-canonical DNA-binding surface on its PHD and wHTH domains. While our identification of two distinct binding sites for the Hop1 CBR on the Widom 601 sequence suggests a degree of sequence specificity, we do not observe any sequence-specific interactions in our structure and can detect no sequence similarity between the two sites (Fig. EV1). These data suggest that binding of the Hop1 CBR to nucleosomes may be driven mainly by recognition of the distinctive bent conformation of DNA in this complex.

We used electrophoretic mobility shift assays (EMSAs) to quantitatively compare binding of the Hop1 CBR to nucleosomes versus a 40-base pair DNA segment encompassing the major Hop1 binding site on the nucleosome. The isolated *S. cerevisiae* Hop1 CBR bound nucleosomes with a $K_d$ of 0.5 μM, compared to 2.1 μM for DNA alone, supporting the idea that the Hop1 CBR specifically recognizes the bent conformation of nucleosomal DNA (Figs. 1F,G and EV2A,B). To test the role of loops 1–3 in nucleosome binding, we generated alanine mutants of the DNA-interacting residues in each loop, plus constructs with two or all three loops mutated. Mutation of loop 1 (N362A/H365A) had the smallest effect on nucleosome binding, reducing the $K_d$ from 0.5 μM to 2.0 μM (Figs. 1G and EV2C). Mutation of loop 3 (K452A/R458A) had a stronger effect, reducing the $K_d$ to ~50 μM, and mutation of loop2 (R402A/K403A/K404A/K405A) had the strongest effect with no detectable binding at the highest protein concentration tested (50 μM) (Figs. 1G and EV2D,E). We also detected no binding when combining the loop2 and 3 mutants, or when combining the loop 1, 2, and 3 mutants (Fig. EV2F–I). Finally, we tested binding of the *S. cerevisiae* Hop1 CBR to nucleosomes with a truncated histone H2A lacking its N-terminal tail, as this tail could theoretically bind the CBR in the observed binding position. In agreement with the lack

of visible density for this tail in our cryo-EM map, we observed no change in binding affinity with these nucleosomes, compared to nucleosomes with full-length H2A (Fig. EV2J). Thus, loop2 is the most important determinant for nucleosome binding by the Hop1 CBR, with loop 3 making a less significant contribution and loop 1 playing only a minor role.

Importantly, all of the above assays were performed with the isolated Hop1 CBR, rather than full-length Hop1. Full-length Hop1 is known to self-interact through HORMA-closure motif binding (West et al, 2018) and to bind Red1 through its N-terminal HORMA domain (Woltering et al, 2000; West et al, 2018), both of which may affect the CBR's accessibility and the overall oligomeric state of Hop1 in cells. Thus, while the CBR is likely the major determinant of nucleosomal binding by Hop1, we cannot rule out direct or indirect contributions from other domains of the protein.

## CBR-nucleosome binding is important for successful meiosis

To investigate the function of CBR-nucleosome binding during meiosis, we mutated loop2 (R402A/K403A/K404A/K405A) in the endogenous *HOP1* locus, creating the *hop1-loop2* allele. The sporulation efficiency of wild type and *hop1-loop2* mutants was comparable (Fig. EV3A), and *hop1-loop2* mutants completed premeiotic DNA replication with wild-type kinetics (Fig. EV3C). Hop1-loop2 protein expression and phosphorylation-dependent mobility shifts also occurred at levels and kinetics like wild type (Fig. EV3D) (Carballo et al, 2008; Subramanian et al, 2016). However, spore viability of *hop1-loop2* mutants was reduced (74.8%) compared to wild type (96.1%), indicating that the nucleosome binding activity of the CBR is important for Hop1 function in vivo (Fig. 2A, $P = 0.0002$). This reduction in spore viability is associated with an increase in 2:2 (live:dead) and 0:4 spore segregation patterns, indicative of a meiosis I defect (Fig. EV3B).

Because Hop1 binding to chromosomes is regulated by the protein remodeler Pch2 (Börner et al, 2008; San-Segundo and Roeder, 1999; Chen et al, 2014; Subramanian et al, 2016; Wojtasz et al, 2009; Mu et al, 2020), we also analyzed *hop1-loop2* mutants in the absence of *PCH2*. *hop1-loop2 pch2Δ* double mutants exhibited premeiotic DNA replication kinetics and sporulation efficiency indistinguishable from wild type (Fig. EV3A,C,D). Hop1 protein levels were elevated in the absence of *PCH2* in agreement with previous work (Fig. EV3D) (Ho and Burgess, 2011; Herruzo et al, 2019). The levels of the mutant Hop1-loop2 protein were similarly elevated, indicating that this effect on Hop1 abundance occurs independently of nucleosome association. Of note, *hop1-loop2, hop1-loop2 pch2Δ*, and *pch2Δ* cells all revealed an additional slow-migrating Hop1 band by western analysis that was not seen in wild type (Fig. EV3D). The biological relevance of this band remains unclear. Strikingly, although *pch2Δ* single mutants exhibited wild-type-like spore viability (95.7%) (San-Segundo and Roeder, 1999) (Fig. 2A), spore viability was synergistically reduced for *hop1-loop2 pch2Δ* double mutants (41.6%, $P = 0.00183$, unpaired *t* test) (Fig. 2A), indicating that in the absence of nucleosome binding by the CBR, cells depend on Pch2 activity for productive meiosis.

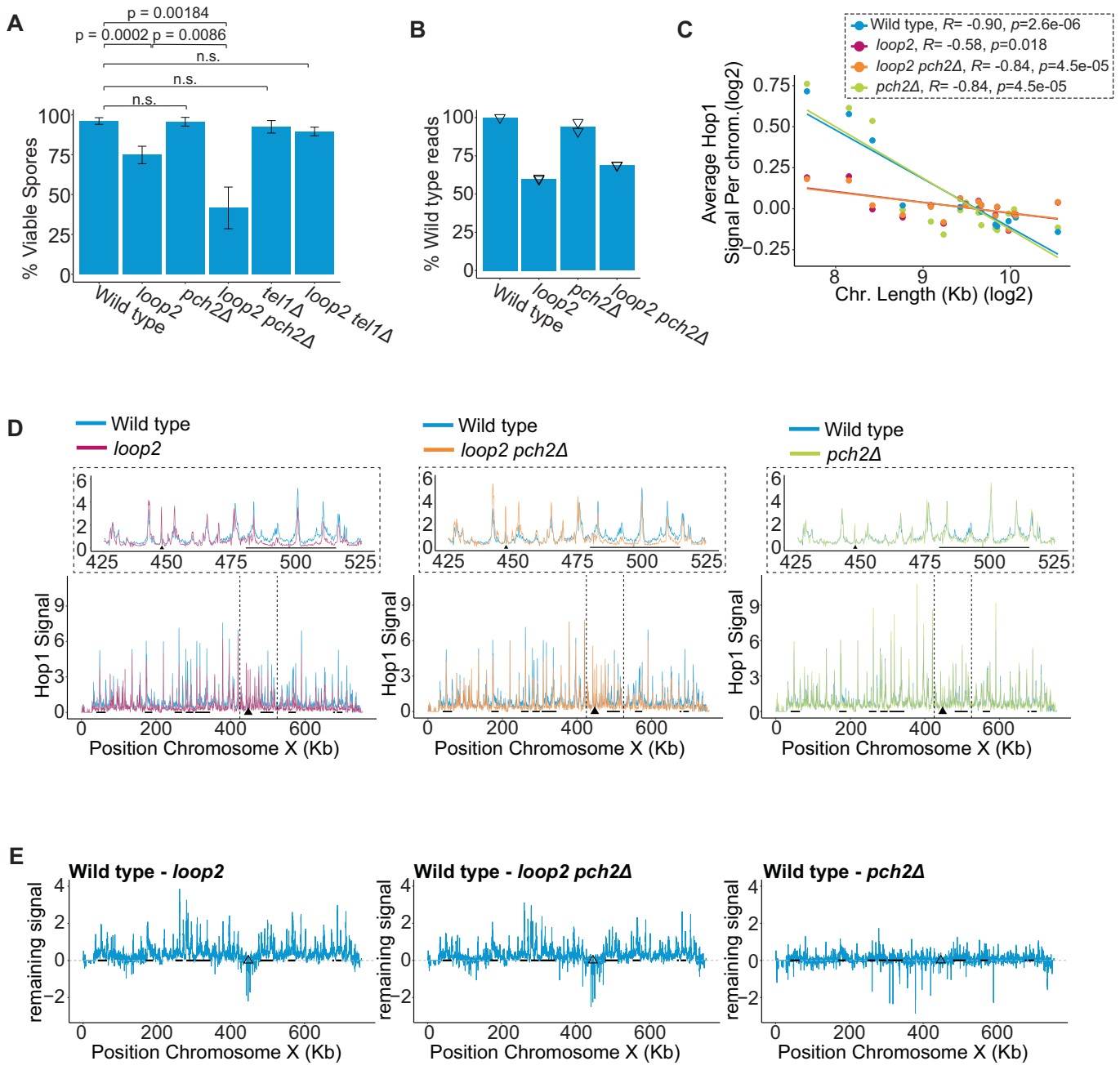

**Figure 2. Hop1 association to DNA is altered in the absence of nucleosome binding.**

The *hop1-loop2* allele is simplified to *loop2* in figure labels. (A) Spore viability for each genotype. *P* values represent the Bonferroni-corrected results of unpaired *t* test between groups indicated by connected lines (*N* > 100 tetrads per genotype). (B) SNP-ChIP normalized average read counts displayed as a percentage of wild-type. Triangles represent values for each replicate (*N* = 2). (C) Mean Hop1 ChIP-seq signal per kb is plotted for each chromosome on a log scale with regression analysis and associated Pearson correlation coefficient for each genotype. (D) Hop1 ChIP-seq signal, normalized by SNP-ChIP is plotted along chromosome X for each genotype. Island regions are underlined with black bars and centromere regions are shown as solid triangles. Dotted lines indicate the region depicted in the zoom panel above each chromosome X plot (*N* = 3). (E) The SNP-ChIP normalized ChIP-sequencing for each genotype is subtracted from wild-type ChIP-seq and then the difference is plotted along chromosome X. All regions above zero indicate regions that contain more Hop1 binding in wild type compared to the corresponding genotype. Island regions are underlined with black bars and centromere regions are shown as open triangles. Source data are available online for this figure.

## Nucleosome–CBR interaction directs Hop1 towards short chromosomes and genomic islands

As the Hop1 CBR specifically promotes axis protein enrichment in island chromatin (Heldrich et al, 2022), we tested whether this

localized enrichment requires nucleosome binding. Spike-in normalized chromatin immunoprecipitation and sequencing (SNP-ChIP normalized ChIP-seq (Vale-Silva et al, 2019)) revealed that total Hop1 chromatin association was reduced to 60% of wild-type in *hop1-loop2* cells and to 69% of wild-type in *hop1-*

**A**

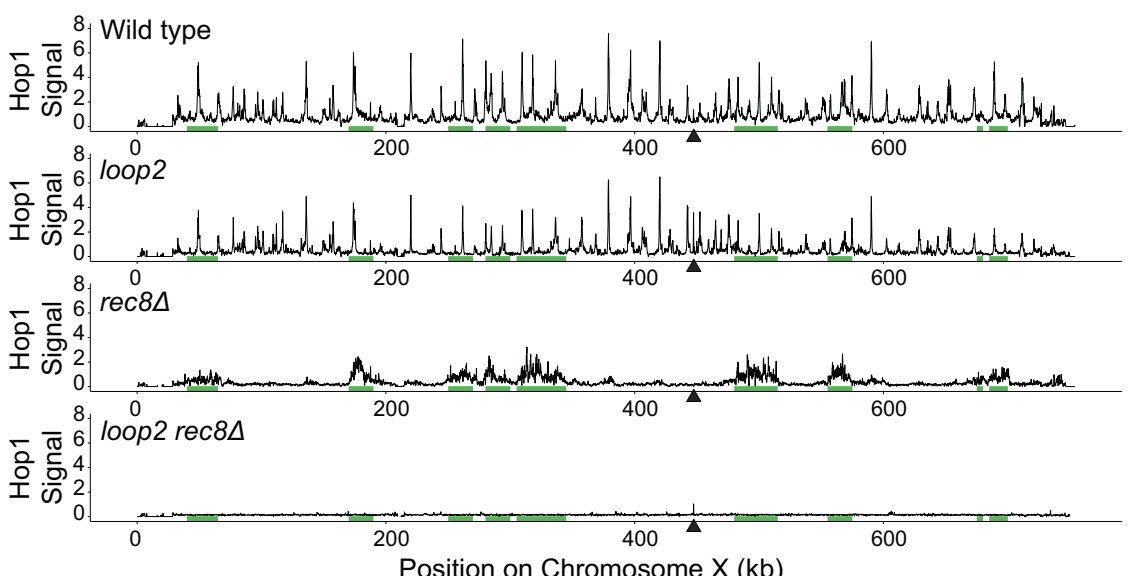

**B**

© The Author(s)

**Figure 3. CBR-nucleosome binding is required for Hop1 enrichment in island regions.**

The *hop1-loop2* allele is simplified to *loop2* in figure labels. (A) Top: Meta-analysis of Hop1 binding in island or non-island regions was compared for each genotype. The solid line represents the average of axis binding, and shaded region corresponds to 95% confidence intervals. Nonoverlapping confidence intervals are considered to be statistically different. Bottom: Hop1 ChIP-seq associated with axis positions is displayed as heatmaps for each genotype, sorted into Island regions or non-island regions and ranked by Hop1 binding strength. (B) Hop1 ChIP-seq signal, normalized by SNP-ChIP is plotted along chromosome X for each genotype. Black triangle represents a centromere. Green bars represent island regions.

*loop2 pch2Δ* double-mutant cells (Fig. 2B), showing that the CBR-nucleosome interaction is necessary for the full chromosomal association of Hop1. Importantly, the loss of Hop1 binding was not uniform across the genome but instead showed several distinct regional effects. In wild-type and *pch2Δ* cells, Hop1 was enriched on the shortest three chromosomes in early prophase (3 h after meiotic induction), but this effect was strongly weakened in *hop1-loop2* and *hop1-loop2 pch2* double mutants (Fig. 2C). Thus, CBR-nucleosome binding helps mediate the relative enrichment of Hop1 on short chromosomes.

Consistent with previous analyses of *hop1* mutants lacking the entire CBR (Heldrich et al, 2022), Hop1 binding was also specifically depleted from island regions in *hop1-loop2* cells and *hop1-loop2 pch2Δ* cells (Fig. 2D; Appendix Fig. S7). By contrast, the *pch2Δ* mutation alone had no effect. These regional differences in Hop1 binding are particularly apparent when the Hop1-loop2 binding profile is subtracted from the wild-type Hop1 binding profile and the differential CBR-dependent enrichment is plotted, which was largely restricted to island regions (Fig. 2E; Appendix Fig. S7).

In wild-type or *pch2Δ* cells, average Hop1 association with island regions was 1.55-fold and 1.52-fold higher than Hop1 association with non-island regions, respectively (Fig. 3A). In contrast, in *hop1-loop2* or *hop1-loop2 pch2Δ* mutants, Hop1-loop2 binding in islands was slightly depleted compared to the binding level in non-island regions (0.88-fold and 0.9-fold, respectively). Heatmap analysis showed that the changes occurred broadly across most, if not all, Hop1 binding sites (Fig. 3A). Spike-in normalization indicated that these relative changes were driven both by reduced binding of Hop1-loop2 in islands and increased binding in non-island regions. Thus, in the absence of nucleosome binding, Hop1 distribution is mildly enriched in non-island sites. This enrichment pattern mirrors the enrichment of Rec8-containing cohesin, which mediates the only remaining axis recruitment mechanism in the absence of CBR activity (Heldrich et al, 2022). Consistently, deletion of *REC8* in *hop1-loop2* mutants caused a complete loss of Hop1 binding (Fig. 3B). We conclude that the cohesin-independent enrichment of Hop1 in island regions requires the direct interaction of the CBR with nucleosomes.

## CBR-nucleosome binding directs Hop1 away from pericentromeric regions

We also analyzed the effect of the *hop1-loop2* mutation on axis enrichment around centromeres. Pericentromeric regions experience different binding patterns among axis proteins. Rec8 is observed to be enriched in pericentromeric regions, whereas Red1 and Hop1 binding in pericentromeric regions is not different from the genome average (Sun et al, 2015; Luo et al, 2023). Spike-in normalized ChIP-seq analysis confirmed that the binding of Hop1

in the centromere-proximal regions ($CEN + / - 20$ kb) in wild-type cells or *pch2Δ* mutant cells was not significantly different from arm regions (Fig. 4A,B). By contrast, in *hop1-loop2* or *hop1-loop2 pch2Δ* cells, Hop1 binding signal in the pericentromeric regions was on average 1.43 and 1.63-fold higher than in arm regions, respectively. This increase was observed for nearly every chromosome and is similar to the binding enrichment observed for Rec8 (Fig. 4B,C), suggesting that Hop1 enrichment in pericentromeric regions is a consequence of Rec8-dependent recruitment. Given that Hop1 is not enriched around centromeres in wild-type cells, these data imply that nucleosome binding by the CBR interferes with Rec8-dependent recruitment of Hop1 in pericentromeric regions.

## Mutation of the Hop1 CBR leads to defective homolog synapsis

To assess Hop1 binding to meiotic chromosomes in individual nuclei, we prepared chromosome spreads from meiotic prophase cells (2, 3, and 4 h after meiotic induction). We probed the chromosomes with antibodies against Hop1 and the SC central-region protein Zip1, a marker of chromosome synapsis (Sym et al, 1993). *hop1-loop2* nuclei in early prophase (hour 2) contained about 26.9% of wild-type Hop1 signal intensity (Fig. 5A,B), and the nuclear area Hop1 occupied (measured as a percentage of total DAPI area) was only about 20% of the nuclear area occupied by wild-type Hop1 (Fig. 5A,C; Appendix Fig. S8). By hour 3, Hop1 signal intensity as well as nuclear area occupied by Hop1 in *hop1-loop2* cells were comparable to wild-type (Fig. 5A). As flow cytometry of the same meiotic cultures indicated that all genotypes entered meiosis synchronously, these results suggest that the association of Hop1 with the axes may be mildly delayed in *hop1-loop2* mutants. Consistent with a defect in Hop1 recruitment, *hop1-loop2 pch2Δ* double mutants exhibited increased Hop1 binding compared to the *hop1-loop2* single mutant, but overall Hop1 binding in the double mutant remained lower than in the *pch2Δ* single mutant (Fig. 5A–C). Moreover, deletion of *PCH2* did not lead to a greater Hop1-loop2 ChIP signal on chromosomes and did not rescue Hop1 binding to island regions (Fig. 2B–D). Thus, while deletion of *PCH2* leads to increased abundance of Hop1-loop2 on chromosomes, it is not sufficient to restore the amount of Hop1-loop2 directly bound to chromatin as assessed by ChIP-seq (Fig. 2). Interestingly, the *hop1-loop2* mutation rescued the reduced DAPI area seen in *pch2Δ* mutants (Appendix Fig. S8), indicating that this phenotype depends on CBR-nucleosome binding.

By late prophase, chromosomal Hop1 levels in wild-type cells declined as a consequence of chromosome synapsis (Subramanian et al, 2016). *hop1-loop2* mutants, however, failed to undergo proper chromosome synapsis and accumulated chromosomal Hop1 like other mutants with synapsis defects (Fig. 5A–C) (Subramanian et al, 2016). To quantify the SC defect of *hop1-loop2* mutants, we

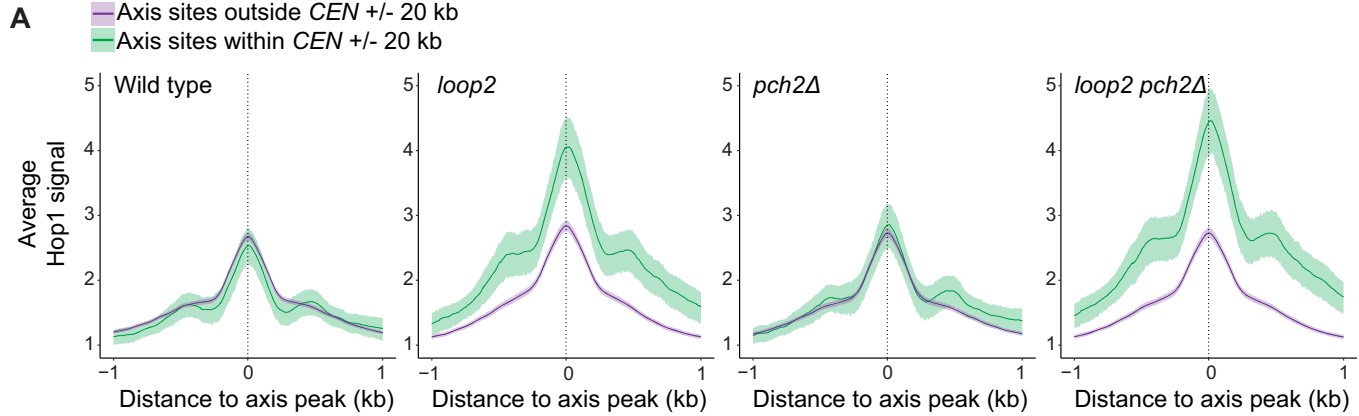

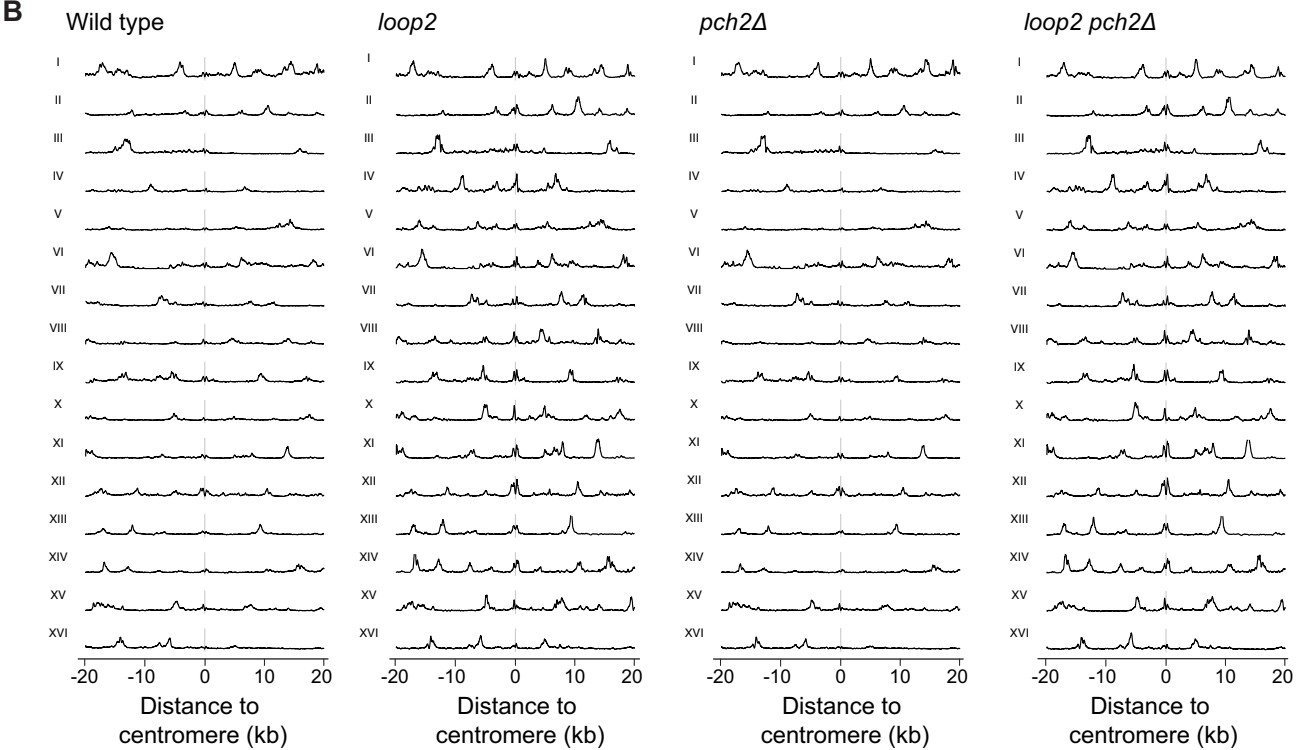

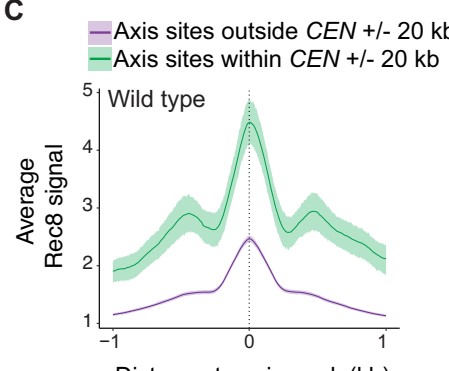

**Figure 4. In the absence of CBR-nucleosome interaction, Hop1 is enriched in centromere regions.**

The *hop1-loop2* allele is simplified to *loop2* in figure labels. (A) Meta-analysis of Hop1 ChIP-seq associated with axis positions in either centromere (defined as any axis site within 20 kb +/− the centromere) or axis positions outside of defined centromere regions. Solid line represents average of axis binding, and shaded region corresponds to 95% confidence intervals. Nonoverlapping confidence intervals are considered to be statistically significant. (B) Hop1 ChIP-seq flanking centromere regions for each chromosome. (C) Same as panel B but for Rec8 ChIP-seq data available for wild type (Sun et al, 2015).

classified nuclei by their Zip1 structures. Whereas the majority of wild-type nuclei displayed full Zip1 polymerization on chromosomes at hours 3 and 4 of meiosis, *hop1-loop2* nuclei displayed primarily punctate Zip1 signals or short Zip1 tracks, similar to Zip1 patterns seen at hour 2 in wild-type (Fig. 5A,E). In addition, *hop1-loop2* nuclei often contained one or two Zip1 aggregates (Fig. 5E). Thus, *hop1-loop2* cells fail to form wild-type SC structures.

Deletion of *PCH2* nearly rescued the SC formation defect of *hop1-loop2* mutants (Fig. 5A). Total Zip1 signal intensity and Zip1 track length in *hop1-loop2 pch2Δ* cells was comparable to that of wild-type cells at hours 2, and 3 after meiotic induction (Fig. 5A,D,E), and chromosomes also showed almost wild-type localization of the SC central-element protein Gmc2 (Humphryes et al, 2013), indicating that *hop1-loop2 pch2Δ* double mutants form bona fide SCs (Fig. EV4). The rescue was incomplete because both Zip1 signal intensity and Zip1 length remained lower than wild-type Zip1 levels at hour 4 (Fig. 5A,D,E). Nonetheless, these results indicate that nucleosome binding by Hop1 is not necessary for meiotic chromosome synapsis. Rather, the defect in SC formation in *hop1-loop2* cells is at least in part a consequence of premature Hop1 removal by Pch2.

### *hop1-loop2* cells experience decreased DSB formation in island regions

To investigate the functional consequences of altered axis assembly in *hop1-loop2* mutants, we assayed DSB induction along whole chromosomes and at individual hotspots by Southern blotting. To exclude differences in DSB turnover, we performed these experiments in a *rad50S* background, which blocks DSB repair (Alani et al, 1990). At the chromosomal level, DSB formation was notably reduced in *hop1-loop2* mutants but still occurred at higher levels than in *hop1Δ* mutants, indicating that Hop1 promotes some DSB formation even in the absence of nucleosome binding (Fig. 6A,B; Appendix Fig. S9). Reduced DSB levels were also observed at individual hotspots (*GAT1*, *ERG1*), although the magnitude of the effect differed depending on the hotspot (Fig. 6C,D; Appendix Fig. S9).

To test whether the Hop1 CBR-nucleosome interaction differentially affects DSB formation in island regions, we mapped DSBs genome-wide by CC-seq (Gittens et al, 2019; preprint: Brown et al, 2023) (Fig. 7A). Consistent with previous analysis (Heldrich et al, 2022), wild-type DSB activity in islands was 1.3-fold that of non-island regions (Fig. 7B). By contrast, in *hop1-loop2* cells, DSB activity in islands was significantly lower than in non-island regions (Fig. 7B). In centromere-proximal regions (*CEN* +/− 20 kb), where Hop1 binding is increased in *hop1-loop2* mutants compared to wild-type (Fig. 4A,B), DSB signals also trended higher although this difference was not statistically significant, likely because of the small sample size (Fig. 7D). Strikingly, short chromosomes were comparatively most affected and showed a notable decrease in

relative DSB activity in *hop1-loop2* mutants (Fig. 7C). These data demonstrate that nucleosome binding recruits Hop1 to specific regions of the genome to alter recombination activity.

### Pch2 promotes DSB formation in *hop1-loop2* cells

If the reduced DSB formation in *hop1-loop2* mutants was solely due to increased Pch2-mediated Hop1 removal from chromosomes, then deletion of *PCH2* should rescue the DSB defect of the *hop1-loop2* mutant. Contrary to this expectation, we found that DSB levels were further reduced in *hop1-loop2 pch2Δ* mutants (Fig. 6A–D; Appendix Fig. S9), explaining the major decrease in spore survival of the double mutant (Fig. 2A). The reduction in DSB levels was observed both by pulsed-field gel electrophoresis and by analysis of individual loci, although we again observed differences in the extent of reduction at individual hotspots. *hop1-loop2 pch2Δ* mutants showed only mildly reduced DSB levels compared to *hop1-loop2* single mutants at the *ERG1* locus, but severely reduced DSB levels at *GAT1* (Fig. 6A–D; Appendix Fig. S9), consistent with locus-specific differences in each hotspot's dependence on Hop1 CBR-nucleosome binding and Pch2. Because synapsis is largely rescued in the *hop1-loop2 pch2Δ* mutants (Fig. 5A,D,E), and because synapsis is known to downregulate DSB induction (Thacker et al, 2014; Subramanian et al, 2016; Mu et al, 2020), we also analyzed *hop1-loop2 pch2Δ zip1Δ* mutants. In triple mutants, DSB formation was not rescued, but was even further reduced than the double mutants (Fig. 6A–D; Appendix Fig. S9). These data indicate that Pch2 is necessary for promoting the DSB activity of Hop1 in the absence of nucleosome binding.

### Nucleosome binding by Hop1 is not necessary for DSB repair template bias

In addition to promoting DSB formation, Hop1 also ensures that meiotic DSB repair uses homologous chromosomes rather than the sister chromatids as the preferred repair template (Niu et al, 2005; Hollingsworth and Byers, 1989). In *hop1Δ* mutants, the few DSBs that do form are largely repaired via the sister chromatid, resulting in a loss of COs (Schwacha and Kleckner, 1994). To investigate the function of CBR-mediated Hop1 recruitment during meiotic DSB repair, we analyzed meiotic recombination at the model *HIS4:LEU2* locus, which allows quantification of CO formation as well as interhomolog and intersister repair by Southern blot analysis (Hunter and Kleckner, 2001). Analysis of the *hop1-loop2* mutant revealed that CO levels were reduced by approximately 50% compared to wild-type (Fig. 8A). This reduction mirrored the 50% reduction in DSB levels at this locus and implies that the DSBs that do form undergo normal CO repair. Consistent with this interpretation, the ratio of joint-molecule repair intermediates between homologous chromosomes and sister chromatids in *hop1-loop2* mutants was near the level of wild-type cells (Fig. 8B). Thus, repair template choice at *HIS4:LEU2*, unlike DSB

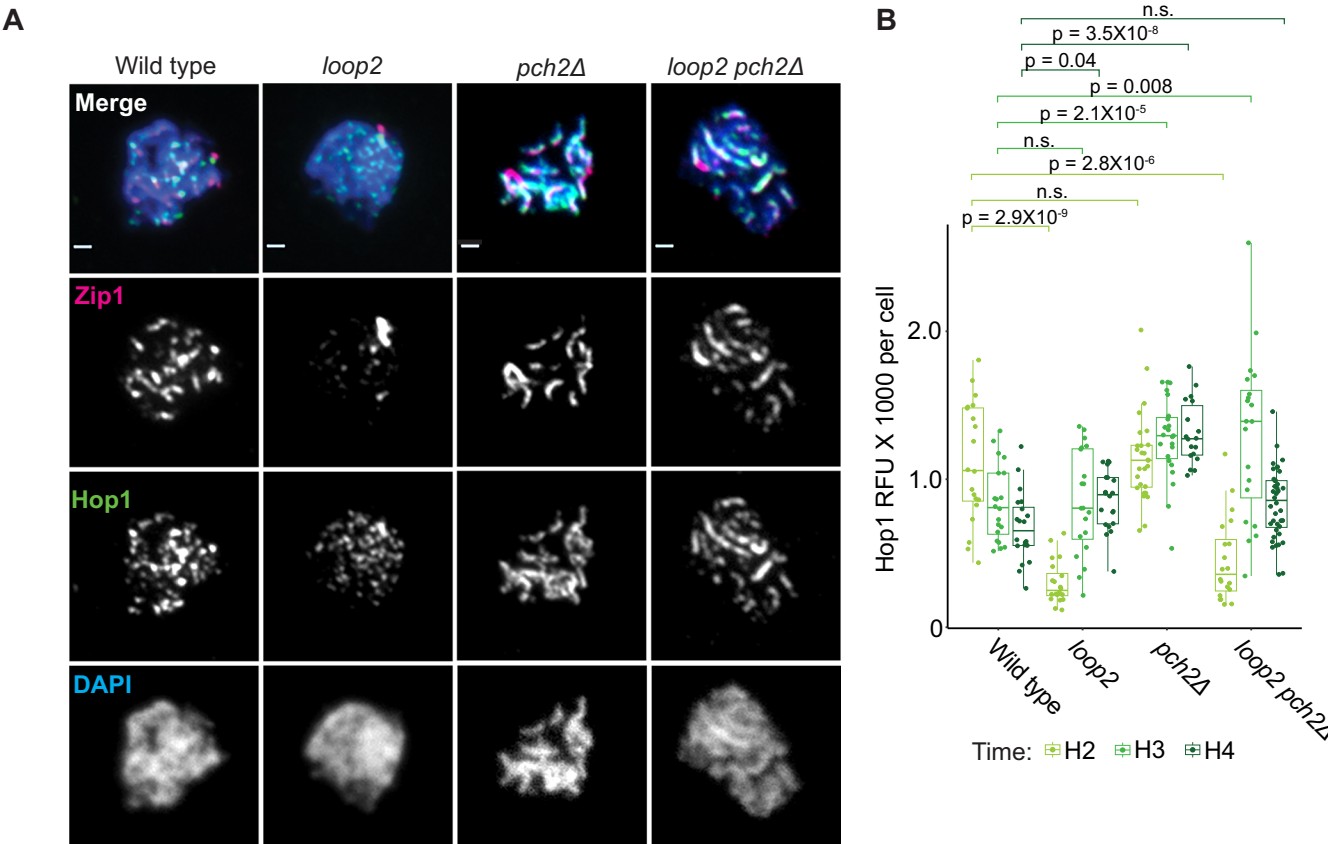

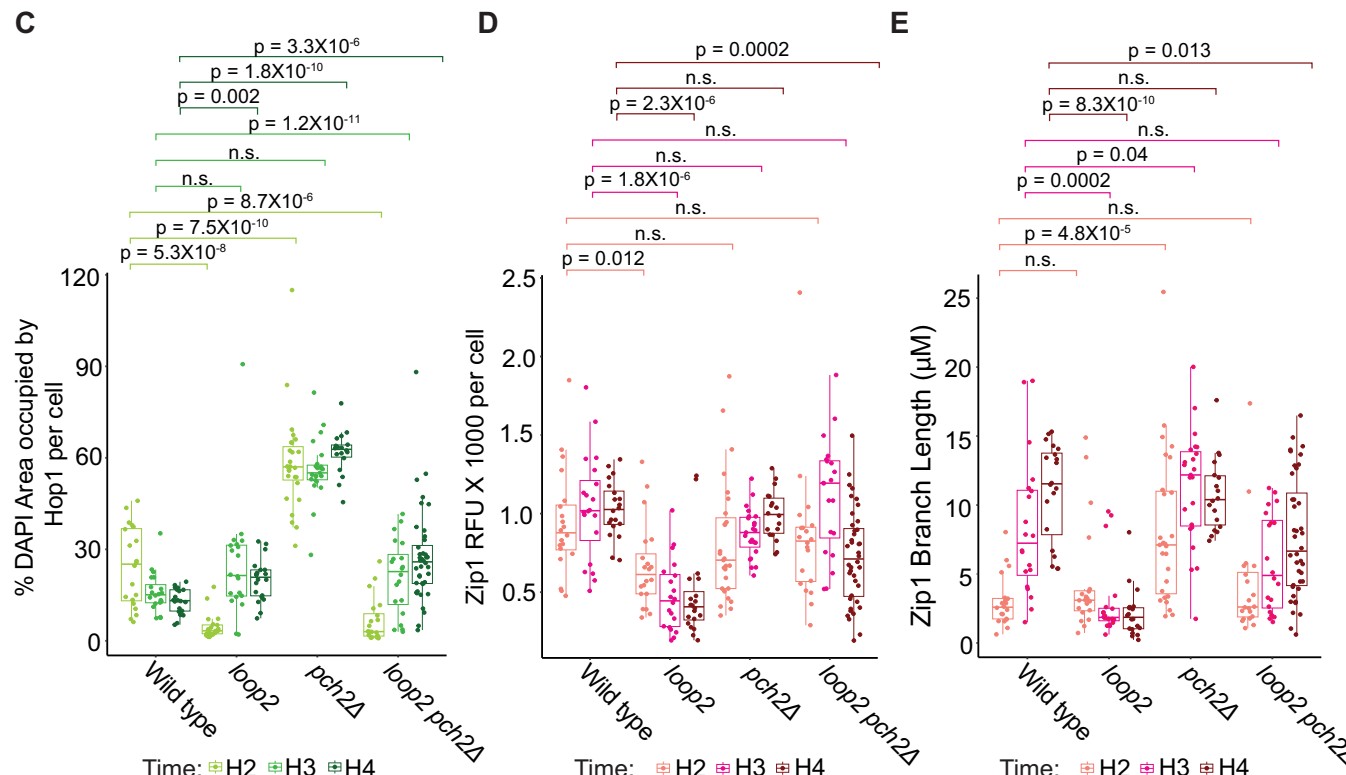

**Figure 5.   Hop1 CBR domain is important for synapsis.**

The *hop1-loop2* allele is simplified to *loop2* in figure labels. (**A**) Representative images from chromosome spreads prepared at 3 h after meiotic induction, hybridized with antibodies against Zip1 and Hop1, and stained with DAPI. Scale bars: 1 μm. (**B**) Dot plot of the quantification of total Hop1 relative fluorescent units (RFU) per cell. Hours 2, 3, and 4 are separated by color as depicted in the legend below. Box plots are overlaid on dot plots and depict interquartile ranges between the 25th and 75th quartile, where the line in the middle of the box represents the medium value. Lines between categories represent *P* value results; n.s. is not significant. Statistics are determined by unpaired Wilcoxon test with Bonferroni correction, $N = 2$. (**C**) Dot plot of the % total DAPI area that the Hop1 signal occupies for each cell. Plot layout, N number, and statistics are the same as (**B**). Individual quantifications of Hop1 and DAPI area for each cell appear in Appendix Fig. S8. (**D**) Dot plot of the quantification of total Zip1 RFU per cell. Plot layout, *N* number, and statistics are the same as (**B**). (**E**) Zip1 length for each cell is displayed in micrometers. Plot layout, *N* number, and statistics are the same as (**B**). Source data are available online for this figure.

formation, does not require nucleosome binding by the CBR. These data functionally separate the CBR from the neighboring SQ/TQ cluster domain (SCD; residues 298–318), which is essential for repair template choice (Carballo et al, 2008). Importantly, the bias toward interhomolog repair remained intact in *hop1-loop2 pch2Δ* double mutants (Fig. 8B), suggesting that the increased spore lethality of the double mutant was primarily due to low DSB numbers and not a consequence of altered repair patterns. Consistent with this interpretation, *TEL1* deletion, which increases the levels of meiotic DSB formation (Garcia et al, 2015; Mohibullah and Keeney, 2017), was sufficient to restore the spore viability of *hop1-loop2* mutants to near wild-type levels (Fig. 2A). Taken together, these data suggest that Hop1-loop2 protein remains fundamentally functional in its ability to support DSB formation, repair bias, and synapsis, and that the primary defect of *hop1-loop2* lies in the regulation of Hop1. We consider potential modes of such regulation in the "Discussion".

## Meiotic HORMADs across eukaryotes possess diverse CBRs

Our data show that the *S. cerevisiae* Hop1 CBR plays important roles in recombination, but this domain is notably lacking in meiotic HORMADs from other model organisms like *C. elegans* and *M. musculus*. To determine how widespread the CBR is among meiotic HORMADs, we generated profile Hidden Markov Models (HMMs) for the PHD, wHTH, and HTH-C domains in *S. cerevisiae* Hop1, then searched for similar domains in meiotic HORMADs from a set of 158 diverse eukaryotes (Tromer et al, 2019; van Hooff et al, 2017). We identified 149 meiotic HORMADs in 158 genomes and transcriptomes representative of the full diversity of the eukaryotic tree of life (thirteen species encode two meiotic HORMADs, and *C. elegans* encodes four). Those organisms that lack an identifiable meiotic HORMAD protein also usually lack homologs of the chromosome axis core proteins (e.g., *S. cerevisiae* Red1 or *M. musculus* SYCP2/SCYP3; (Tromer et al, 2021)). The meiotic HORMADs we could identify almost always encode both an N-terminal HORMA domain and a C-terminal closure motif, but show a strikingly variable architecture within their central regions as detected by HMM profile matches and AlphaFold structure predictions (Fig. 9A,B; Table EV1) (Varadi et al, 2022; Jumper et al, 2021). The majority of HORMADs we identified (107 of 149) encode a CBR containing some combination of a PHD and a wHTH domain. Across Saccharomycetaceae—the fungal group that contains *S. cerevisiae*—we detected a conserved CBR architecture with PHD, wHTH, and HTH-C domains. In a large set (36/149) of meiotic HORMADs from other Opisthokonta (including Fungi and Metazoa), the CBR comprises predicted PHD and wHTH domains, but lacks a clear HTH-C (Fig. 9A,B; Appendix Fig. S10). In these proteins, the PHD domain's canonical histone tail binding hydrophobic cage is highly conserved

(Appendix Fig. S10A,B), and the wHTH domain's canonical DNA-binding site is predicted to be surface-exposed and positively charged (Appendix Fig. S10C–E). These data suggest that, unlike *S. cerevisiae* Hop1, the larger group of CBRs lacking HTH-C may bind nucleosomes in a bipartite manner through both histone tails and DNA. In the pathogenic fungi *Encephalitozoon cuniculi* and *Encephalitozoon intestinalis*, we identified putative meiotic HORMADs that lack an N-terminal HORMA domain, and encode only a CBR with a PHD/wHTH domain or PHD domain, respectively (Fig. 9A,B; Table EV1).

In addition to CBRs encoding PHD and wHTH domains, a large set of meiotic HORMADs encode CBRs with either a single wHTH domain (46/149, scattered throughout the tree) or a tandem repeat of two such domains (25/149; in *Mantamonas*, some Amoebozoa, SAR, and Archaeplastida) (Figs. 9A,B and EV5; Table EV1). Notably, both *Arabidopsis thaliana* ASY1 and *Oryza sativa* PAIR2 encode two wHTH domains. These domains may bind DNA in a sequence-specific or nonspecific manner to promote meiotic chromosome axis assembly similarly to CBRs with PHD and wHTH domains.

Based on the conserved domains within the meiotic HORMAD central region, we propose that the last common ancestor of eukaryotes encoded a meiotic HORMAD protein with one or two wHTH domains. In Opisthokonta (Holozoa+Holomycota), this architecture was elaborated to include a PHD domain situated N-terminal to a single wHTH domain (wHTH is most similar to the C-terminal wHTH of double wHTH HORMADs), then further extended in Saccharomycetaceae to include the HTH-C domain (Fig. 9C). The addition of the HTH-C domain also apparently coincided with the development of the distinct DNA-binding surface we observe by cryo-EM. Intriguingly, we also observe yet other chromatin-binding domains (Fig. 9C; Table EV1), such as a putative methyl-binding PWWP-like domain (Chlorophyta, Cryptista) or unknown domains (*Aurantiochytrium*). Many organisms (30/133), notably including mammals and most nematodes, have since lost the CBR entirely, and their meiotic HORMADs apparently encode only an N-terminal HORMA domain and a disordered C-terminal region with a closure motif at the protein's extreme C-terminus (Fig. 9A). Strikingly, however, the majority of meiotic HORMADs across eukaryotes encode a CBR with predicted DNA and/or chromatin-binding activity.

## Discussion

The meiotic chromosome axis promotes faithful chromosome inheritance and fertility by regulating DSB formation and repair across the genome. The mechanisms that modulate the genomic distribution and activity of meiotic axis proteins remain poorly understood. We have shown here that the Hop1 CBR binds to

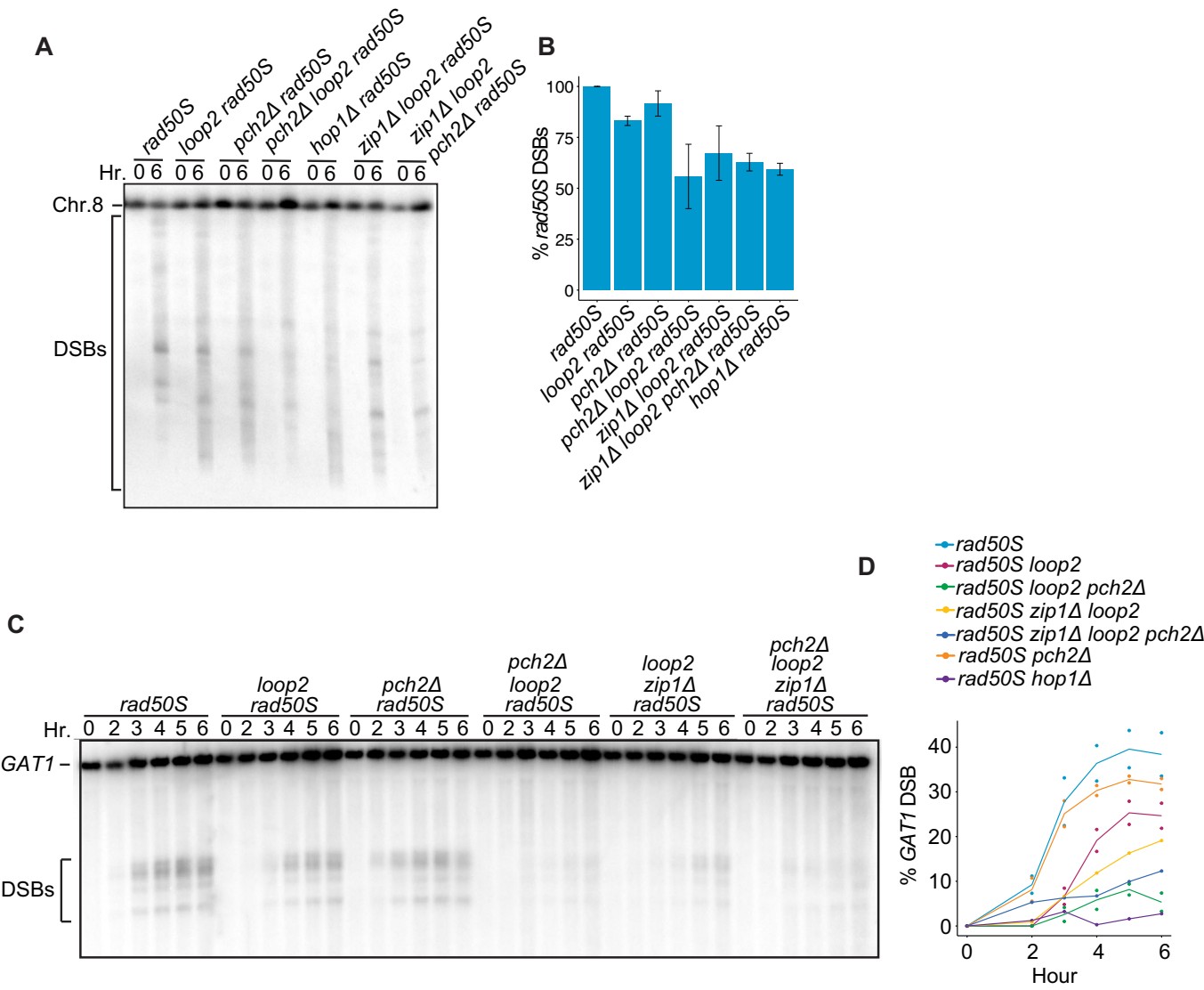

**Figure 6. Hop1 CBR domain promotes DSB activity.**

The *hop1-loop2* allele is simplified to *loop2* in figure labels. (A) Pulsed-field gel and Southern blot analysis of chromosome VIII (analyzed using a probe against *CBP2*). (B) Percent total broken DNA calculated based on a Poisson distribution using the intact chromosome VIII bands for each genotype and assuming full replication by 6 h (Thacker et al, 2014). Data represents four biological replicates, error bars indicate standard deviation. (C) Southern blot analysis of the *GAT1* locus across genotypes during meiosis. The *GAT1* hotspot is located outside an island on chromosome 6 (the second smallest chromosome). Top band represents the intact *GAT1* locus. Bottom band represents DSBs that occur at the *GAT1* locus. (D) Measurements of the DSB induction for each genotype at the *GAT1* locus. Data represents two biological replicates for each genotype, with the exception of *zip1Δ hop1-loop2* and *zip1Δ hop1-loop2 pch2Δ*, which were analyzed once. Source data are available online for this figure.

nucleosomes in vitro, and functions in vivo to drive Hop1 binding to short chromosomes and gene-dense genomic islands. We find that nucleosome binding is important for Hop1 function in initiating recombination and SC formation. Both functions are highly dependent on the Hop1 protein remodeler Pch2, implying a role for the CBR in controlling Hop1 turnover.

## Nucleosome binding by the CBR patterns axis protein distribution and DSB formation

Our data show that the CBR-nucleosome interaction contributes to axis patterning as well as to the DSB landscape by directing Hop1

binding towards island chromatin. The resulting Hop1 enrichment in islands functions to increase the frequency of recombination locally, an activity that is particularly important for short chromosomes because it helps promote the CO formation necessary for faithful segregation during meiosis I (Kaback et al, 1992; Chen et al, 2008; Kaback, 1996; Mancera et al, 2008).

The CBR-nucleosome interaction independently recruits Hop1/Red1 to chromatin, acting in parallel with Rec8 to drive the final axis protein binding. Initial Hop1/Red1 recruitment, which occurs concurrently with premeiotic S phase (Blitzblau et al, 2012) is likely driven largely by Rec8 because ChIP-seq analysis shows a binding enrichment of Hop1 around centromeres at early time points (hour

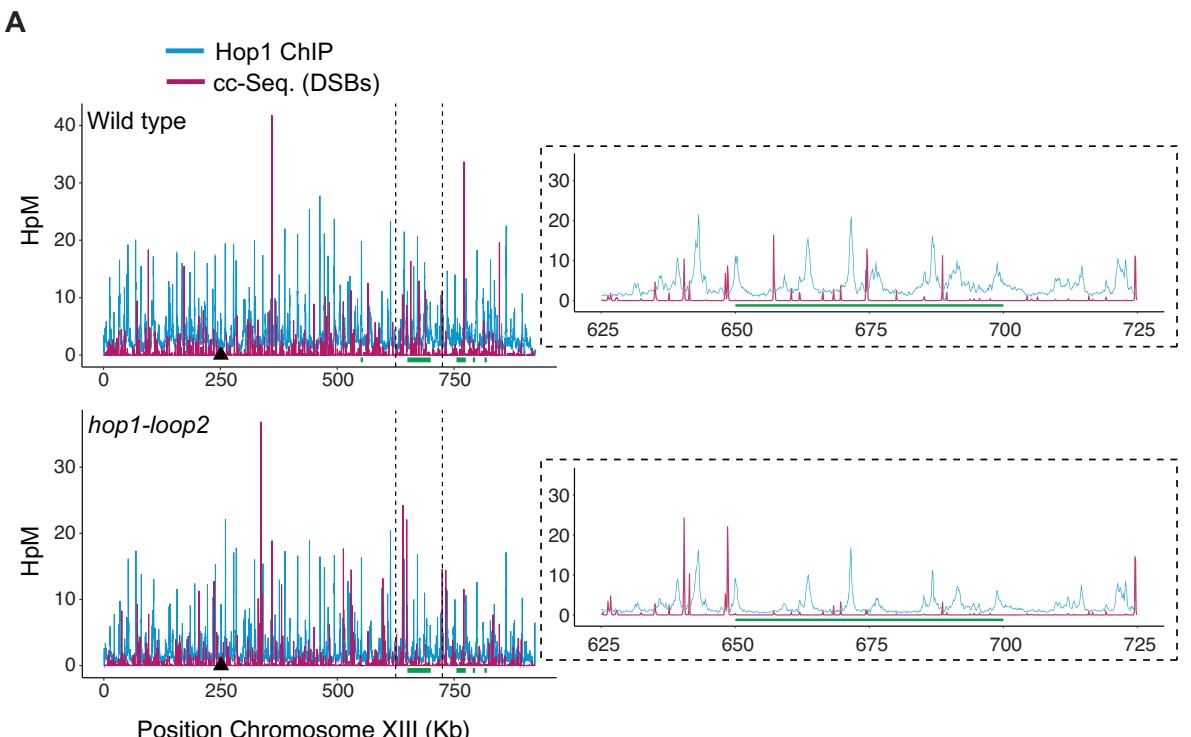

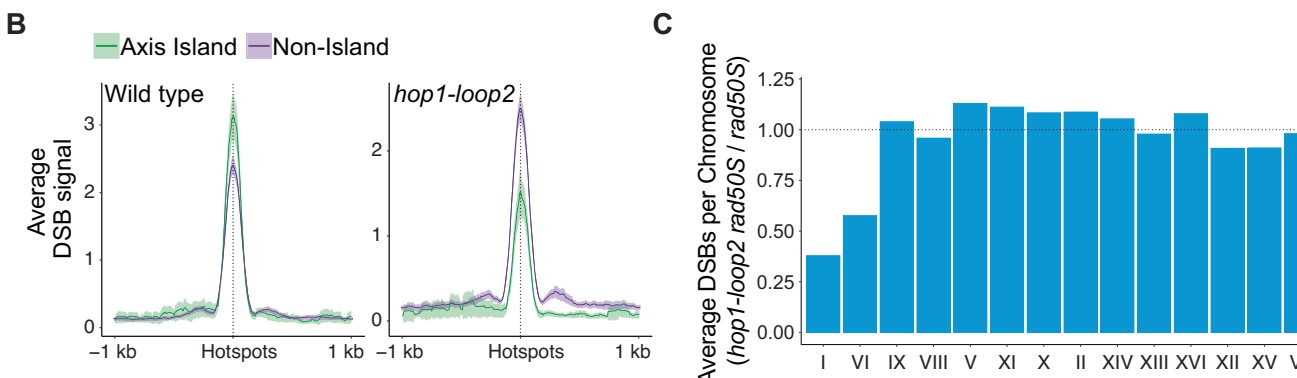

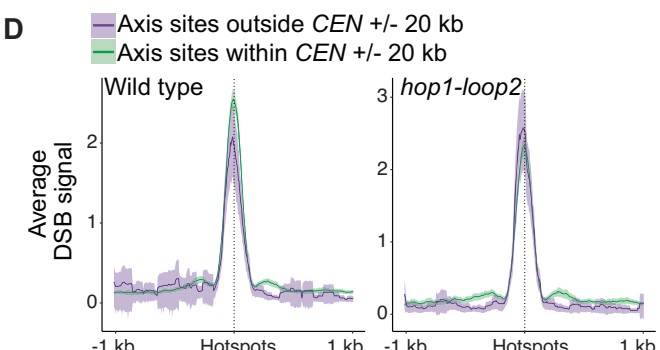

**Figure 7.   Genome-wide DSB analysis reveals altered Hop1 binding in the absence of CBR-nucleosome interaction directly affects DSB distribution.**

(A) CC-sequencing (magenta), with two replicates, is plotted along chromosome XIII as hits per million (HpM) for *rad50S* and *rad50S hop1-loop2* cultures collected at 6 h. Hop1 ChIP signal (blue) is also plotted to compare the relative positions of Hop1 binding and DSB activity. The dotted lines indicate the zoom region plotted to the right of the whole chromosome plot. Black bars represent island regions, black triangles represent centromeres. (B) Meta-analysis of CC-sequencing data comparing read counts associated with hotspots in axis island regions with read counts associated with hotspots in non-axis regions for either *rad50S* cells or *rad50S hop1-loop*2 cells. Shaded regions outside of the averages represent 95% confidence intervals. (C) Ratio plots for the average number of DSBs per chromosome in *hop1-loop2 rad50S* compared to the average number of DSBs per chromosome in *rad50S* cultures. The chromosomes are organized from smallest to largest. (D) Similar plots as (B), except comparing CC-sequencing data associated in hotspots 20 kb +/− the centromere to CC-sequencing data associated with hotspots outside these regions.

2) (Subramanian et al, 2019). This early Hop1/Red1 enrichment is more like the pericentromeric enrichment observed of Rec8 binding than like the final distribution pattern of Hop1 or Red1, which does not show binding enrichment in centromeres. It is possible that Hop1/Red1 molecules are initially recruited to chromatin by Rec8, and the CBR acts to redistribute the pool of Hop1/Red1 molecules. It is equally plausible that the CBR domain drives an independent population of Hop1/Red1 molecules to chromatin, thereby reducing the pool of molecules available for Rec8 recruitment, a model that could explain the reduced Hop1-loop2 binding revealed by SNP-ChIP. These two possibilities are not mutually exclusive.

The CBR-nucleosome interaction specifically drives Hop1/Red1 binding to island chromatin, as demonstrated by Hop1 binding in *rec8Δ* mutants (Sun et al, 2015; Heldrich et al, 2022). What feature of island chromatin the CBR-nucleosome interaction distinguishes from the rest of the genome remains an open question. Nucleosomes are not unique to island chromatin, and therefore the CBR-nucleosome interaction alone cannot drive the specificity for island regions. Previous investigations have failed to reveal sequence motifs or chromatin modifications unique to island regions (Sun et al, 2015), and although DNA sequences underlying islands do have a higher-than-average propensity to bind nucleosomes, this feature is also not unique to islands (Heldrich et al, 2022). The relative non-specificity of the CBR-nucleosome interaction is supported by our structural analyses, as CBR-nucleosome binding occurs primarily through interactions with the DNA backbone. Furthermore, while our results found the CBR preferentially binds to specific positions along the nucleosomal Widom 601 DNA sequence, this specificity likely reflects an affinity for certain bent conformations of DNA. It is possible the interaction of loop 1 with the major groove has some sequence specificity, but this contribution is likely limited because disruption of loop 1 had only minor effects on binding affinity. The CBR also showed no preference for modified histones or histone tails in vitro, but instead revealed a preference for unmodified histones. Additional factors must therefore exist to specify Hop1 enrichment in island regions.

One candidate driver for island specificity is local chromosome context. Recently, long-range effects of centromeres and telomeres have been recognized to act on axis protein recruitment to short chromosomes, and these may also explain the abundance of islands on short chromosomes (Murakami et al, 2020; Subramanian et al, 2019; Luo et al, 2023). It is also possible that the propensity for Hop1 to form higher-order assemblies with additional Hop1/Red1 units (West et al, 2019, 2018) could create a complex with multiple CBR domains, which could cooperate in cis to increase Hop1 binding locally. Such cooperative effects are consistent with the higher enrichment of Hop1 (per kb) observed on larger islands compared to smaller islands

(Heldrich et al, 2022). Although long-range chromosome effects and higher-order assemblies of Hop1/Red1 complexes may play a role in driving CBR specificity to islands, there is no obvious molecular mechanism, indicating that additional factors remain to be identified to explain island specificity.

## CBR-nucleosome interaction controls meiotic DSB formation and synapsis via Pch2 regulation

Hop1 supports meiotic recombination at multiple levels, including SC assembly, DSB initiation, and CO repair. Loss of the CBR-nucleosome interaction leads to lower Hop1-loop2 protein bound to chromosomes and selectively impairs SC assembly and DSB initiation. Unlike mutants lacking *HOP1* entirely, *hop1-loop2* mutants execute repair template bias as expected at *HIS4:LEU2*, and although we cannot exclude defects at other loci, these data indicate the Hop1-loop2 protein is competent in establishing the block of sister repair and promoting CO repair. In addition, Hop1-loop2 protein levels and phosphorylation levels are normal, and the recruitment of Hop1-loop2 by Rec8 seems largely unaffected. These results indicate that *hop1-loop2* is not hypomorphic but instead is a separation-of-function allele.

The SC defect observed in *hop1-loop2* mutants was unexpected. Hop1 is known to be necessary for SC assembly (Hollingsworth and Byers, 1989), but to our knowledge, no other *hop1* allele results in a comparable partial SC assembly phenotype. The possibility that the Hop1-loop2 protein is somehow defective in recruiting or assembling components of the SC was considered, but we do not favor this explanation because SC assembly is rescued upon deletion of *PCH2*. These results indicate that the Hop1-loop2 protein can support SC assembly, but that the regulation by Pch2 prevents normal SC assembly.

One model consistent with these phenotypes is that Pch2 more readily extracts Hop1-loop2 from chromosomes and thus limits the amount of synapsis-promoting Hop1 on chromosomes. However, if premature removal of Hop1-loop2 by Pch2 was the only defect, then deletion of *PCH2* should have rescued the DSB defect of *hop1-loop2* mutants. Yet, despite abundant Hop1-loop2 localization upon deletion of *PCH2*, *hop1-loop2 pch2Δ* double mutants incurred fewer DSBs than *hop1-loop2* mutants. In principle, the further reduction of DSB formation observed among *hop1-loop2 pch2Δ* double mutants could be due to the elevated synapsis in these mutants, as synapsis is known to downregulate DSB formation (Thacker et al, 2014; Subramanian et al, 2016; Mu et al, 2020). However, this explanation is unlikely because DSB downregulation is mediated by the Pch2-dependent removal of Hop1 (Subramanian et al, 2019), and this pathway cannot be active in *hop1-loop2 pch2Δ* mutants. Accordingly, eliminating the SC (*zip1Δ*) did not rescue the

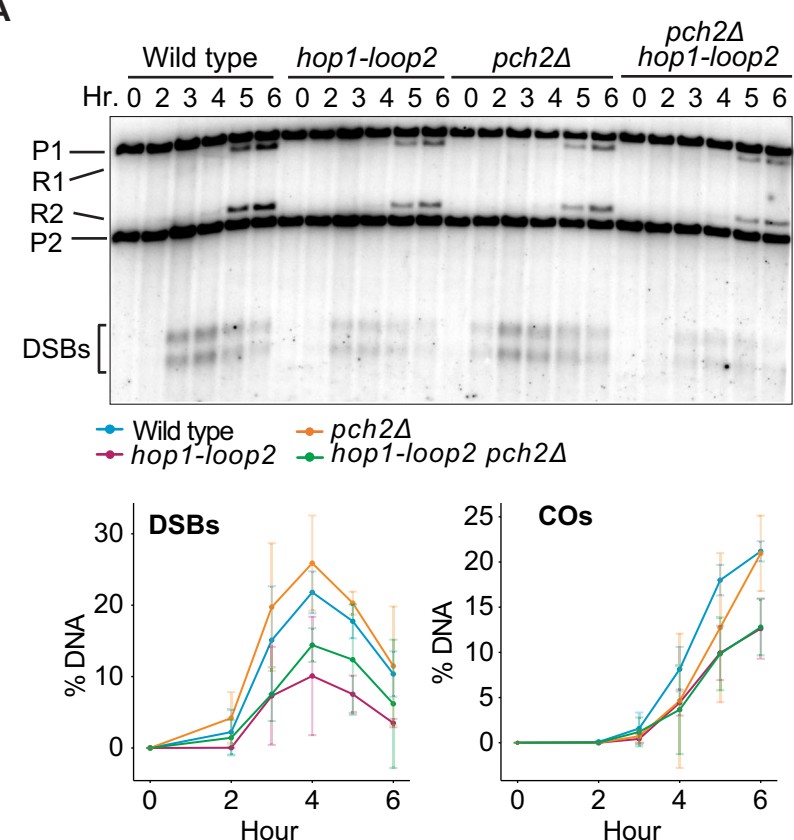

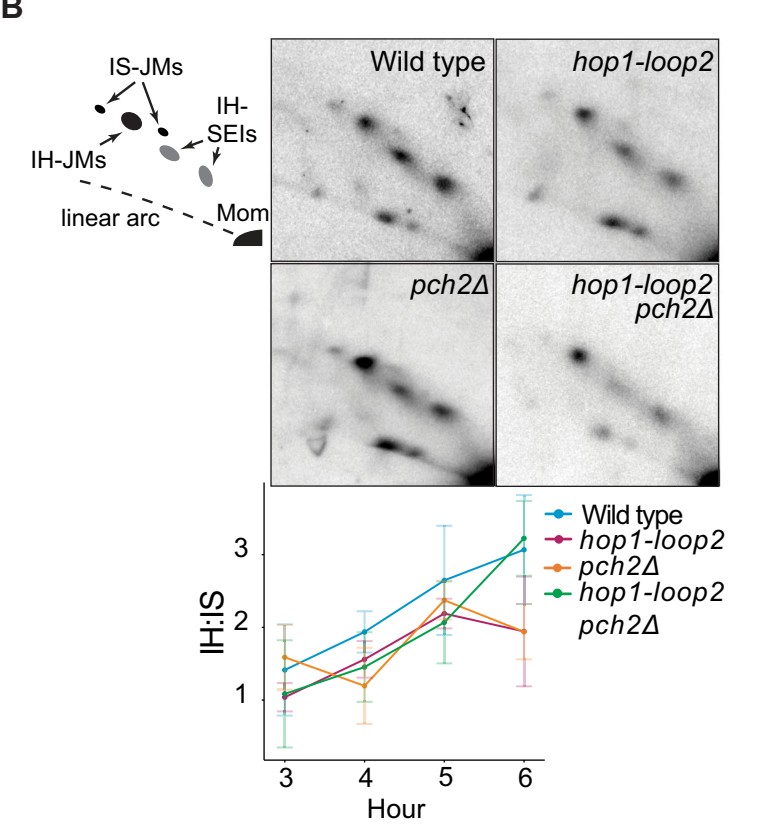

**Figure 8. Hop1 CBR domain does not impact repair choice.**

(A) Top: Southern blot analysis of the *HIS4:LEU2* hotspot locus across genotypes during meiosis. P1 and P2 represent parental alleles. R1 and R2 represent COs. DSBs appear near the bottom of the gel. Bottom left: Quantification of DSBs from three replicates at the *HIS4:LEU2* locus, from three replicates, shown as a percentage of the total DNA signal for each lane; error bars indicate standard deviation. Bottom right: Measurements of recombination between P1 and P2 are depicted as a percentage of total DNA; error bars indicate standard deviation. (B) Native-native 2D gel electrophoresis and Southern blot analysis of psoralen-stabilized recombination intermediates at the *HIS4:LEU2* locus. Top left: Illustration of the expected pattern of recombination intermediates. IS-JM intersister join molecules, IH-JM interhomolog joint molecules. SEIs single-end invasion intermediates, a precursor of IH-JMs. Top right: representative image for each genotype at hour 4. Bottom: Measurements, from three replicates, of joint-molecule signals depicted as the ratio of IH-JM signal to IS-JM signal; error bars indicate standard deviation. Source data are available online for this figure.

DSB defect of these mutants. These opposite effects of *PCH2* on DSB formation and synapsis in *hop1-loop2* mutants indicate that these two processes have a different dependence on Hop1 turnover.

To explain the observed phenotypes, we propose that chromosomal Hop1 exists in two states (Fig. 10). One state, which predominates in early prophase and which we refer to as "state 1", promotes DSB formation and is refractory to Pch2-dependent disassembly. The second state, "state 2" is unable to promote DSB formation but supports synapsis and is susceptible to disassembly by Pch2. Once in state 2, Hop1 is unable to return to state 1 without the intervention of Pch2 activity. In this model, the CBR-nucleosome interaction stabilizes Hop1 in state 1 (Fig. 10). When the CBR domain is unbound, as in *hop1-loop2* mutants, Hop1 more readily transitions from state 1 to state 2, enabling untimely Pch2-mediated removal from chromatin (Fig. 10). We propose that recycling of Hop1-loop2 by Pch2 is responsible for the diminished SC assembly in *hop1-loop2* mutants but also restores Hop1-loop2 protein to state 1 to promote further DSB formation. By contrast, in *hop1-loop2 pch2Δ* double mutants, Hop1 is not recycled back to the DSB-promoting state 1, resulting in the accumulation of Hop1-loop2 protein in state 2, which permits extensive Zip1 deposition but prevents further DSB formation.

The idea that Hop1 exists in multiple states is consistent with available data. Hop1 is extensively modified by phosphorylation and SUMOylation, which may impact its susceptibility to Pch2-mediated disassembly (Penedos et al, 2015; Carballo et al, 2008; Bhagwat et al, 2021). In addition, structural studies of HORMA proteins predict that Hop1 participates in different protein-protein interactions by engaging its HORMA domain with closure motifs in Red1 and possibly other Hop1 monomers to form multi-protein complexes (West et al, 2018). A single Hop1 monomer can likely also form a "self-closed" state, in which its HORMA domain binds to its own C-terminal closure motif. Pch2 is predicted to regulate Hop1 states by pulling apart these interactions (West et al, 2018) and thus aiding in the interconversion of Hop1 states. Studies of Hop1 in the meiotic checkpoint, as well as studies of the related Mad2 protein in the mitotic spindle assembly checkpoint, suggest that these different conformational states have different functions (Raina and Vader, 2020; Raina et al, 2023).

The existence of two different states of Hop1, that are differentially susceptible to disassembly by Pch2, may also help explain the well-known anticorrelation between Hop1 and chromosome synapsis. In many organisms, including budding yeast, mouse, and *Arabidopsis*, HORMA proteins are enriched on unsynapsed axial elements but are depleted on synapsed chromosomes (Yang et al, 2022; Wojtasz et al, 2009; Börner et al, 2008). PCH2/TRIP13 localizes specifically to synapsed chromosomes (San-Segundo and Roeder, 1999; preprint: Chotiner et al, 2023; Balboni et al, 2020) and thus is well positioned to remove HORMA proteins. However, PCH2 is also expressed and present in the nucleus prior to synapsis (Yang et al, 2022; preprint: Chotiner et al, 2023; Herruzo et al, 2019) and yet, does not seem to remove Hop1 at this point, which has led to the proposal that Hop1 is refractory to removal prior to synapsis (Yang et al, 2022). Based on these observations, we speculate that the two states of Hop1 may align temporally with chromosome synapsis, with state 1 (DSB-active and Pch2-resistant) corresponding to axial Hop1 prior to synapsis and state 2 (DSB-inactive and subject to Pch2 remodeling) occurring in the context of, and possibly because of, synapsis. As such, the ability of Hop1 to bind to nucleosomes, stabilizing Hop1 in a DSB-active state, may be integral to its ability to regulate the progression of meiotic recombination.

### The CBR is broadly conserved among eukaryotic HORMAD proteins

Across eukaryotes, meiotic HORMADs possess CBRs of variable architecture: fungi and many animals encode a CBR with a tandem PHD and wHTH domain while most other eukaryotes encode CBRs with one or two wHTH domains, with some having replaced the wHTH with other chromatin interaction domains like PWWP and others losing the CBR altogether. Thus, meiotic HORMADs share a broadly conserved module capable of direct chromatin binding, which can complement cohesin-mediated axis assembly and play a key regulatory role in meiotic recombination.

## Methods

### Protein expression and purification

Ligation-independent cloning was used to clone *S. cerevisiae* Hop1$^{322-537}$ and *V. polyspora* Hop1$^{319-529}$ into UC Berkeley Macrolab vectors to generate N-terminal TEV protease-cleavable His$_6$- or His$_6$-maltose binding protein tags (Addgene #29666, 29706). DNA-binding mutants of *S. cerevisiae* Hop1$^{319-529}$ (loop 1: N362A + H365A; loop2: R402A + K403A + K404A + K405A ; loop 3: K452A + R458A) were generated by PCR mutagenesis of His$_6$-MBP tagged constructs.

For protein expression, plasmids encoding Hop1 fragments were transformed into *E. coli* LOBSTR cells (Kerafast), and grown in 2XYT media supplemented with 10 μM ZnCl$_2$. Cells were grown at 37 °C to an OD$_{600}$ of 0.6, then protein expression induced with 0.25 mM IPTG and cells were grown a further 16 h at 20 °C.

For protein purification, cells were harvested by centrifugation, suspended in resuspension buffer (20 mM Tris-HCl pH 7.5, 300 mM NaCl, 20 mM imidazole, 2 mM β-mercaptoethanol, and 10% glycerol) and lysed by sonication. The lysate was clarified by

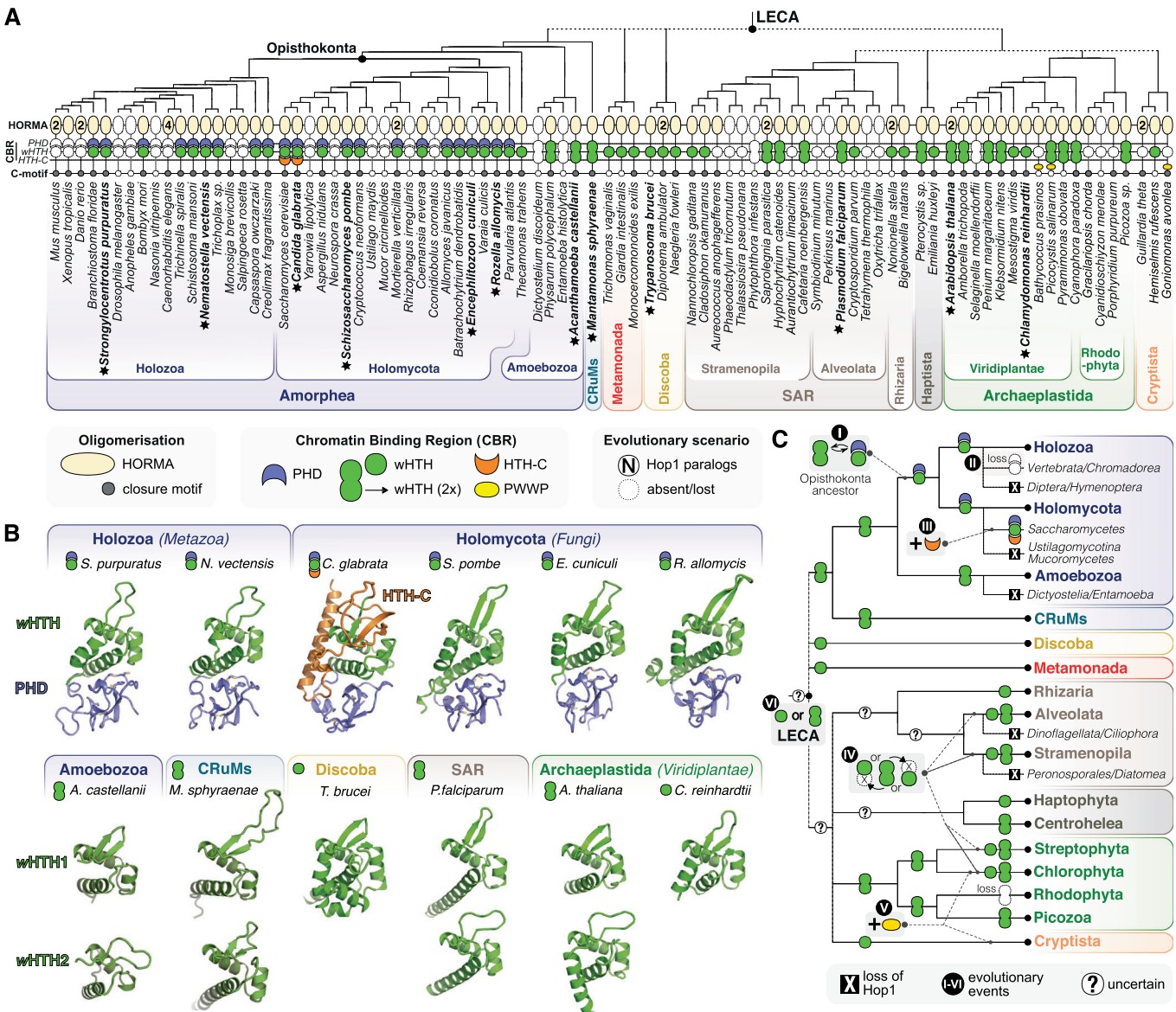

**Figure 9.  Meiotic HORMADs encode a variable central chromatin-binding region.**

(A) Cartoon of a phylogenetic tree of diverse eukaryotes (a subset of 87 out of 158 species analyzed in this study), with individual domains in meiotic HORMAD proteins noted: HORMA domain beige, PHD domain blue, wHTH domain green, HTH-C domain orange, PWWP yellow and the closure motif grey. Numbers in the HORMA domain field indicate the number of detected meiotic HORMADs in a given genome. Two different shapes are depicted for the wHTH domain, each indicating the number of wHTH domains detected in a single meiotic HORMAD protein. Major eukaryotic groups are color-coded and labeled. See full dataset of 158 species/149 HORMAD proteins in Table EV1. (B) AlphaFold 2 predicted structures of selected meiotic HORMAD CBRs (indicated with a star and in bold text in (A)). All CBRs are shown with PHD domain blue, wHTH green, and HTH-C orange. *Strongylocentrotus purpuratus* (Uniprot ID A0A7M7T0L9) residues 329–476 (PHD 329–395, wHTH 396–476); *Nematostella vectensis* (Uniprot ID A7RLI6) residues 280–426 (PHD 280–345, wHTH 346–426); *Candida glabrata* (Uniprot ID Q6FIP6) residues 319–526 (PHD 319–373, wHTH 374–438, HTH-C 439–526); *Schizosaccharomyces pombe* (Uniprot ID Q9P7P2) residues 333–480 (PHD 333–388, wHTH 389–480); *Encephalitozoon cuniculi* (Uniprot ID Q8SWC8) resides 1–142 (full length; PHD 1–61, wHTH 62–142); *Rozella allomycis* (Uniprot ID A0A075AU51) residues 399–551 (PHD 399–454, wHTH 455–551); *Acanthamoeba castellanii* (Uniprot ID L8H1D9) residues 246–309 (wHTH 1) and 335–394 (wHTH 2); *Mantamonas sphyraenae* (ID see Table EV1) residues 254–340 (wHTH 1) and 3787–465 (wHTH 2); *Trypanosome brucei* (Uniprot ID Q38B55) residues 292–406 (wHTH); *Plasmodium falciparum* (Uniprot ID Q8IEM0) residues 973–1053 (wHTH 1) and 1214–1287 (wHTH 2); *A. thaliana* ASY1 (Uniprot ID F4HRV8) residues 297–362 (wHTH 1) and 388–471 (wHTH 2); *Chlamydomonas reinhardtii* (Uniprot ID A0A2K3DBA0) residues 443–516 (wHTH). Of note: *C. reinhardtii* wHTH is most similar to *A. thaliana* wHTH 1, while *T. brucei* wHTH is more similar to other wHTH 2 domains. See Fig. EV5 for sequence alignments. (C) Reconstruction of the evolution of the CBR of meiotic HORMAD protein throughout the eukaryotic tree of life. Six key events can be discerned: (I) replacement of the N-terminal wHTH domain in the ancestor of fungi and animals (Opisthokonta) with a PHD domain; (II) loss of the PHD-wHTH configuration in a number of animal lineages (i.e., Vertebrata and the nematode class Chromadorea); (III) C-terminal extension (HTH-C) of the wHTH specific for *Saccharomycetes*; (IV) differential loss of one of the two wHTH domains in various clades within the supergroups SAR and Archaeplastida; (V) independent gains of PWWP-like domains in Chloroplastida and Cryptista; (VI) unclear (inferred) ancestral presence of a single or double wHTH configuration in the Last Eukaryotic Common Ancestor (LECA). Source data are available online for this figure.

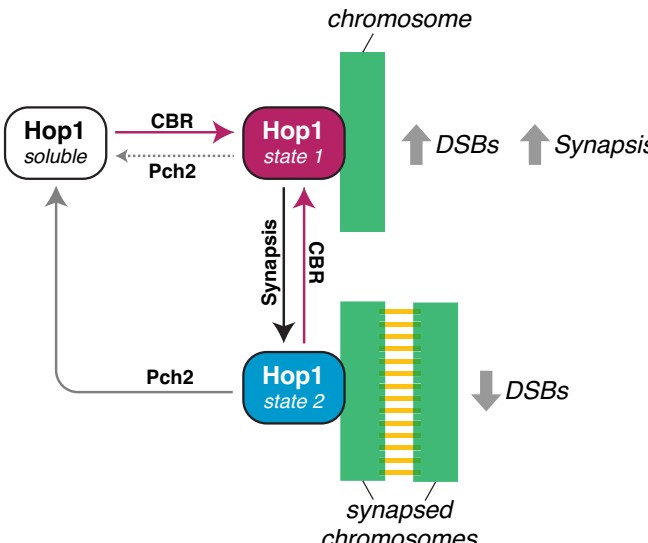

**Figure 10. The CBR-nucleosome interaction regulates Hop1 function.**

We propose that Hop1 exists in at least three states: a soluble state primed by Pch2 for chromatin association (white), and two chromatin-associated states (State 1 purple, State 2 blue). State 1 predominates upon initial chromatin association and promotes DSB formation, which in turn promotes synapsis. State 2, which does not promote DSB formation, predominates after synapsis. Pch2 can readily access and remodel Hop1 in state 2, but state 1 is refractory to Pch2. The Hop1 CBR prevents Hop1 remodeling by stabilizing Hop1 in state 1, preventing the switch to state 2. In the absence of CBR-nucleosome binding, a premature switch from state 1 to state 2 occurs. This switch results in lower DSB levels and premature conversion of Hop1 into the soluble form, destabilizing the SC. Some recombination activity is salvaged by Pch2-mediated recycling Hop1. In the absence of both CBR-nucleosome binding domain and Pch2, Hop1 accumulates in state 2 without recycling, resulting in further decreased DSB formation while permitting SC assembly.

centrifugation, then the supernatant was loaded onto a $Ni^{2+}$ affinity column (HisTrap HP, Cytiva) pre-equilibrated with resuspension buffer. The column was washed with resuspension buffer, followed by low-salt wash buffer (20 mM Tris-HCl pH 7.5, 100 mM NaCl, 20 mM imidazole, 2 mM β-mercaptoethanol, and 10% glycerol) and eluted with low-salt wash buffer supplemented with 250 mM imidazole. The eluate was loaded onto an anion-exchange column (Hitrap Q HP, Cytiva) and eluted with a gradient to high-salt wash buffer (20 mM Tris-HCl pH 7.5, 1 M NaCl, 20 mM imidazole, 2 mM β-mercaptoethanol, and 10% glycerol). Fractions containing desired protein were pooled and concentrated to 500 µL by ultrafiltration (Amicon Ultra-15, EMD Millipore), then passed over a size exclusion column (HiLoad Superdex 200 PG, Cytiva) in size exclusion buffer (20 mM Tris-HCl pH 7.5, 300 mM NaCl, 10% glycerol, and 1 mM DTT). For crystallization experiments, the N-terminal His$_6$-tag on *V. polyspora* Hop1[317-535] was cleaved by TEV protease treatment (16 h at 4 °C) of pooled anion-exchange fractions. The cleavage reaction was passed over a second $Ni^{2+}$ affinity column and the flow-through containing cleaved protein was collected, concentrated, and further purified by size exclusion. Purified proteins were concentrated by ultrafiltration and stored at 4 °C for crystallization, or aliquoted and frozen at −80 °C for biochemical assays. All mutant proteins were purified as wild-type.

## Nucleosome core particle reconstitution

Lyophilized *Xenopus laevis* histone proteins (H2A, H2B, H3, and H4) were purchased from the Histone Source at Colorado State University (https://histonesource-colostate.nbsstore.net). Histone octamer and full nucleosome assembly was performed essentially as described (Luger et al, 1999). Briefly, histone octamers were assembled by resuspending individual purified histones in unfolding buffer (6 M Guanidinium HCl, 20 mM Tris-HCl pH 7.5, 5 mM DTT) to a concentration of 2 mg/mL, then mixing in a molar ratio of 1 H2A:1 H2B:0.9 H3:0.9 H4 and diluted to a final concentration of 1 mg/mL. The mixture was placed in a dialysis cassette (Pierce Slide-A-Lyzer, 3.5 kDa MWCO) and dialyzed three times (4 h, 12 h, 4 h) into refolding buffer (2 M NaCl, 10 mM Tris-HCl pH 7.6, 1 mM EDTA, 1 mM DTT). Assembled octamers were separated from unassembled H2A:H2B dimer by size exclusion (Superdex 200 PG) in refolding buffer.

A 146-bp DNA fragment corresponding to the Widom 601 DNA sequence (Lowary and Widom, 1998) (sequence below) was amplified by PCR and purified by anion-exchange chromatography (HiTrap Q HP, Cytiva). DNA and histone octamers were assembled by overnight dialysis from 1.4 M TEK buffer (1.4 M KCl, 10 mM Tris-HCl pH 7.5, 0.1 mM EDTA, 1 mM DTT), to 10 mM TEK buffer (10 mM KCl, 10 mM Tris-HCl pH 7.5, 0.1 mM EDTA, 1 mM DTT). Fully assembled nucleosomes were purified by size exclusion (Superose 6 10/300 GL, Cytiva) to ensure quality and purity.

>Widom 601 DNA sequence (chain J in models; reverse complement is chain I in models).

TGGAGAATCCCGGTGCCGAGGCCGCTCAATTGGTCGTA-
GACAGCTCTAGCACCGCTTAAACGCACGTACGCGCTGTCC
CCCGCGTTTTAACCGCCAAGGGGATTACTCCCTAGTCTCC
AGGCACGTGTCAGATATATACATCCTGT.

## X-ray crystallography

For crystallization, *V. polyspora* Hop1[317-535] was exchanged into crystallization buffer (20 mM Tris-HCl pH 7.5, 50 mM NaCl, 1 mM DTT) and concentrated to 20 mg/ml. Crystals were grown by mixing concentrated protein 1:1 with well solution containing 0.1 M MES pH 6.5 and 30% (v/v) PEG 400, in hanging-drop format. A single-wavelength anomalous diffraction (SAD) dataset at the zinc K-edge (wavelength 1.283 Å) was collected at ALS beamline 12.3.1 (Appendix Table S1). Data were indexed and integrated by XDS (Kabsch, 2010), then scaled and converted to structure factors by AIMLESS and TRUNCATE (Kabsch, 2010; Winn et al, 2011). For structure determination, two anomalous sites representing protein-bound zinc ions were identified by hkl2map/ SHELX (Sheldrick, 2010) and input into the Phenix Autosol pipeline (Liebschner et al, 2019) for phasing and automatic model building. An initial model was manually rebuilt in COOT (Emsley et al, 2010), and refined in phenix.refine (Afonine et al, 2012; Pattersen et al, 2021). Figures were generated with PyMOL (version 2.0; Schrödinger, LLC) or ChimeraX (Pettersen et al, 2021).

## Cryoelectron microscopy sample preparation

For cryo-EM sample preparation, reconstituted nucleosomes were incubated with a 20-fold molar excess of *S. cerevisiae* Hop1[322-537] in cross-linking buffer (20 mM HEPES 7.5, 50 mM NaCl) for 30 min on

ice, brought to room temperature for 5 min, then cross-linked with a final concentration of 0.02% glutaraldehyde for 30 min at room temperature. The cross-linking reaction was quenched using 1 volume of 1 M glycine, concentrated and passed over a Superose 6 size exclusion column (Cytiva) (Appendix Fig. S3B), then concentrated again to 3 μM. 3.5 μL of sample was applied to a Quantifoil Cu 1.2/1.3 300 grid (Electron Microscopy Sciences), then blotted and frozen after a 1 min incubation using a Vitrobot Mark IV (Thermo Fisher) with blot force 20, blot time 5.5 s, 4 °C, and 100% humidity.

## Cryo-EM data collection and refinement

A full dataset of 1314 micrographs were collected using a FEI Titan Krios G3 at magnification 130,000× (1.1 Å per pixel), using a K2 Summit direct electron detector, at a defocus range of −0.5 to −2 μm. cryoSPARC (Structura Biotechnology) version 3 was used for patch motion correction, patch CTF estimation, and particle picking (blob picker; 1,267,442 initial particles; Appendix Fig. S4). Multiple rounds of 2D class averaging resulted in a refined set of 373,963 particles, which were re-extracted and subjected to ab initio 3D reconstruction with three classes. One class (94,297 particles) showed one Hop1 CBR bound to a nucleosome, and a second class (114,228 particles) showed a nucleosome without Hop1 bound. Both sets of particles were subjected to heterogeneous refinement, yielding final maps at 2.74 Å resolution (Hop1 CBR:nucleosome complex) and 2.57 Å (nucleosome alone) (Appendix Table S3). The Hop1 CBR:nucleosome complex was also subjected to local refinement using a mask covering Hop1 CBR, to a resolution of 3.15 Å (Appendix Fig. S4). This map was used to manually rebuild a threaded model of _S. cerevisiae_ Hop1[322-524] generated by the PHYRE2 server (Kelley and Sternberg, 2009) based on the structure of _V. polyspora_ Hop1[319-529]. For the nucleosome core particle, PDB ID 3LZ0 (Vasudevan et al, 2010) was used as a template for rebuilding. Models were manually rebuilt in COOT (Emsley et al, 2010) using cryo-EM maps sharpened using DeepEMhancer (Sanchez-Garcia et al, 2021), then subjected to real-space refinement in phenix.refine (Afonine et al, 2012). To choose the correct DNA sequence orientation, both orientations of the Widom 601 sequence were built and refined, then the Pearson's correlation coefficient between model and map were calculated for each base pair, then averaged over each orientation, and the higher-correlation orientation was used.

## Biochemical assays

For initial binding assays with _S. cerevisiae_ Hop1[322-537] and nucleosomes, biotinylated _H. sapiens_ nucleosomes were purchased from Epicypher (product # 16-0006). Ten μg of _S. cerevisiae_ Hop1[322-537] was mixed with 10 μg nucleosomes in binding buffer (20 mM Tris-HCl pH 7.5, 2 mM β-mercaptoethanol, 25 mM NaCl, 10% glycerol) in 50 μL total reaction volume, and incubated at room temperature for 30 min. 10 μL of streptavidin beads (Dynabeads MyOne Streptavidin T1; Thermo Fisher) were added and the mixture incubated for a further 30 min at room temperature with agitation. Beads were washed 3× with 1 mL of binding buffer, and bound proteins were eluted with SDS-PAGE loading dye, boiled, and analyzed by SDS-PAGE.

For electrophoretic mobility shift assays (EMSA), 50 μL reactions were prepared in binding buffer with nucleosome (or DNA) concentration held constant at 50 nM and varying the

concentration of _S. cerevisiae_ Hop1[322-537]. After 15 min room temperature incubation, sucrose was added to a final concentration of 5% (w/v), and reactions were separated on 6% TBE-acrylamide gels pre-equilibrated (pre-run for 60 min at 150 V) in 0.2X TBE running buffer. Gels were run for 90 min at 120 Volts at 4 °C. Gels were stained with Sytox Green and visualized using a Bio-Rad ChemiDoc system. Gel bands were quantified in ImageJ (Schneider et al, 2012), and binding curves were calculated using GraphPad Prism version 9 (GraphPad Software).

>40-base DNA from Widom 601 sequence:
GAGGCCGCTCAATTGGTCGTAGACAGCTCTAGCACCG.

## Yeast genetics, growth, and meiotic time-course assays

_S. cerevisiae_ strains were generated with PCR-based methods (Longtine et al, 1998) or CRISPR-based mutagenesis (DiCarlo et al, 2013). The strains used in this study are listed in Appendix Table S4. For spore viability, cells were grown on YPD agar and then patched onto SPO medium (1% KOAc) for 48–72 h. At least 100 tetrads (400 spores) were dissected for each strain. For synchronous meiosis, cells were grown in YPD, then diluted into BYTA (1% yeast extract, 2% bactotryptone, 1% potassium acetate, 50 mM potassium phthalate) at an $OD_{600}$ of 0.3, grown overnight, then washed and resuspended in SPO medium (0.3% potassium acetate pH 7.0) at an $OD_{600}$ of 2.0 at 30 °C to induce sporulation. Samples were taken at hours 0, 2, 3, 4, 5, 6, 7, 8, and processed. Synchrony of all time courses was verified by flow cytometry; cells are fixed in ethanol, treated with proteinase K and RNAse prior to staining with SYTOX Green DNA staining, and analyzed DNA content using BD FACSAria.

## ChIP-sequencing

Chromatin immunoprecipitation sequencing (ChIP-seq) samples were prepared from 25 mL of meiotic cultures, 3 h after meiotic induction. Each sample was immediately cross-linked with 1% formaldehyde for 30 min at room temperature and then the formaldehyde was quenched by adding glycine to a final concentration of 125 mM. Fixed cells were collected by centrifugation at 2000 RPM for 3 min, the supernatant was removed, and cell pellets were immediately frozen at −80 °C. until the ChIP-seq protocol is continued. Cell pellets were resuspended in 500 μL of lysis buffer (50 mM HEPES/KOH pH 7.5, 140 mM NaCl, 1 mM EDTA, 1% Triton X-100, 0.1% sodium deoxycholate) with protease inhibitors (1 mM PMSF, 1 mM Benzamidine, 1 mg/ml Bacitracin, one Roche Tablet (catalog # 11836170001) in 10 ml) and glass beads in a biopulveriser. Samples were sonicated at 15% amplitude for 15 s 5× to obtain DNA at an average length of 500 bp. Sonicated cell lysate was centrifuged at 14,000 RPM for 10 min at 4 °C to remove cell debris, and the supernatant containing soluble chromatin fraction was isolated. In total, 50 μL of the chromatin fraction was set aside from each sample to use as input samples. To the remaining chromatin fraction, we added 2 μL of anti-Hop1 antibody (a kind gift from Nancy Hollingsworth) and incubated rotating overnight at 4 °C. Antibodies and associated DNA were then isolated using Gammabind G Sepharose beads (GE Healthcare Bio, catalog # 17-0885-01). Crosslinks were reversed by incubating samples and input at 65 °C for at least 6 h. Proteins and RNAs were removed using proteinase K and RNaseA. Libraries for ChIP-sequencing were prepared by PCR amplification using Illumina

TruSeq DNA sample preparation kits v1 and v2. Libraries were quality checked on 2200 Tapestation. Libraries were quantified using Qubit analysis prior to pooling. The ChIP libraries were sequenced on Illumina NextSeq 500 instruments at the NYU Biology Genomics core to yield 150 bp paired-end reads. For spike-in normalization (SNP-ChIP), SK288c cross-linked meiotic samples were added to respective samples at 20% prior to ChIP processing (Vale-Silva et al, 2019). SNP-ChIP libraries were sequenced on NextSeq 500 to yield 150 bp paired-end reads. All ChIP-seq datasets are listed in Table EV2.

## Covalent-complex sequencing (CC-seq) mapping

Protein-DNA Covalent-Complex Mapping (CC-seq) in yeast followed a method previously described (Gittens et al, 2019; preprint: Brown et al, 2023). Briefly, meiotic cell samples were chilled and frozen at −20 °C for at least 8 h, then thawed and spheroplasted (in 1 M sorbitol, 50 mM NaHPO4, 10 mM EDTA, 30 min at 37 °C), fixed in 70% ice-cold ethanol, collected by centrifugation, dried briefly, then lysed in STE (2% SDS, 0.5 M Tris, 10 mM EDTA). Genomic DNA was extracted via Phenol/Chloroform/IAA extraction (25:24:1 ratio) at room temperature, with aqueous material carefully collected, precipitated with ethanol, washed, dried, then resuspended in 1× TE buffer (10 mM Tris/1 mM EDTA). Total genomic DNA was sonicated to <500 bp average length using a Covaris M220 before equilibrating to a final concentration of 0.3 M NaCl, 0.2% Triton X-100, 0.1% Sarkosyl. Covalent complexes were enriched on silica columns (Qiagen) via centrifugation, washed with TEN solution (10 mM Tris/1 mM EDTA/0.3 M NaCl), before eluted with TES buffer (10 mM Tris/1 mM EDTA/0.5% SDS). Samples were treated with Proteinase K at 50 °C, and purified by ethanol precipitation. DNA ends were filled and repaired using NEB Ultra II end-repair module, with custom dsDNA adapters ligated sequentially to the sonicated, then blocked, ends with rSAP and then recombinant TDP2 treatment in between these steps to remove the 5-phosphotyrosyl-linked Spo11 peptide. Ampure bead cleanups were used to facilitate sequential reactions. PCR-amplified libraries were quantified on a Bioanalyser and appropriately diluted and multiplexed for deep sequencing (Illumina NextSeq1000 2×51 bp). For further information, a detailed protocol is available (preprint: Brown et al, 2023).

FASTQ reads were aligned to the reference genome (SK1Spike; which includes the 2-micron plasmid from SacCer3, Lambda (J02459.1) and Human (hg19) inserted into the SK1 S. cerevisiae genome build (Yue et al, 2017) via Bowtie2, using TermMapper as previously described (Gittens et al, 2019; preprint: Brown et al, 2023) (https://github.com/Neale-Lab/terminalMapper), with all subsequent analyses performed in R version 4.1.2 using RStudio (Version 2021.09.0 Build 351). Reproducibility between libraries for independent biological replicates, and quality control, was evaluated and validated prior to averaging libraries using public scripts (https://github.com/Neale-Lab/CCTools). Details of individual libraries are presented in Appendix Table S7.

## Processing of reads from Illumina sequencing for ChIP-seq analyses

Illumina output reads were processed in the following manner. The reads were mapped to the SK1 genome (GCA_002057885.1) using

Bowtie (Yue et al, 2017). SNP-ChIP library reads were aligned to concatenated genome assemblies of SK1 and S288c genomes (Vale-Silva et al, 2019). Only reads that matched perfectly to the reference genome were retrieved for further analysis. 3' ends of the reads were extended to a final length of 200 bp using MACS2 2.1.1 (https://github.com/taoliu/MACS) and probabilistically determined PCR duplicates were removed. The input and ChIP pileups were SPMR-normalized (single per million reads) and fold-enrichment of ChIP over input data was used for further analysis. The pipeline to process Illumina reads can be found at https://github.com/hochwagenlab/ChIPseq_functions/tree/master/ChIPseq_Pipeline_v4. The pipeline used to process SNP-ChIP reads and calculate spike-in normalization factor can be found at https://github.com/hochwagenlab/ChIPseq_functions/tree/master/ChIPseq_Pipeline_hybrid_genome.

ChIP-seq datasets were made up of at least two biological replicates that were merged prior to using the ChIPseq_Pipeline_v4. All datasets were normalized to the global mean of one and regional enrichment was calculated. The R functions used can be found at https://github.com/hochwagenlab/hwglabr2 and analysis and graphs can be found at https://github.com/hochwagenlab/Hop1-loop2.

Processed and raw data files can be found in the GEO repository: GSE225129.

## Chromosome spreads

Meiotic cells were collected at hour 2, 3, and 4 after meiotic induction and treated with 200 mM Tris pH 7.5/20 mM DTT for 2 min at room temperature and then spheroplasted in 2% potassium acetate/1 M Sorbitol/ 0.13 μg/μL zymolyase T100 at 30 °C. The spheroplasts were rinsed and resuspended in ice-cold 0.1 M MES pH 6.4/1 mM EDTA/0.5 mM MgCl₂/1 M Sorbitol. Cells were placed on glass slide and one volume of fixative (1% paraformaldehyde/3.4% sucrose/0.14% Triton X-100) was added to the cells on a clean glass slide (soaked in ethanol and air-dried) and distributed across the slide by gentle tilting. Four volumes of 1% lipsol were added to the slide and mixed by tilting. Cells were spread by a clean glass rod. Four volumes of fixative solution were added and slides were left to dry overnight and stored at −80 °C the next day. Prior to staining, slides were washed in 1× PBS/0.4% Kodak Photoflo, and blocked with 1× PBS/1% chicken egg white albumin (Sigma Aldrich #A5503) on a shaker for 15 min at room temperature. Slides were hybridized with antibodies at 37 °C for 1 h (antibodies and corresponding dilutions are indicated in Appendix Table S6). Slides were then washed with 1× PBS/0.05% Triton-X three times for 5 min each. Secondary antibodies were hybridized at 37 °C for 1 h (Appendix Table S6). Slides were washed once with 1× PBS/0.05% Triton-X, once with 1× PBS/0.4% Kodak Photoflo, and twice with dH₂0/0.4% Kodak Photoflo and air-dried prior to mounting with VECTASIELD with DAPI mounting medium (VWR #H-1200-10) and a VWR Micro Cover Glass, No. 1; 60 × 24 mm (VWR #48393 106). Stained slides were stored at 4 °C prior to imaging.

## Microscopy and cytological analysis

Images were collected on a Deltavision Elite imaging system (GE) equipped with an Olympus 100×/1.40 NA UPLSAPO PSF oil immersion lens and an InsightSSI Solid State Illumination module. Images were captured using an Evolve 512 EMCCD camera in the conventional mode and analyzed using ImageJ software. ImageJ

analysis scripts can be found at https://github.com/hochwagenlab/Hop1-loop2. Scatterplots were generated using the GGplot2 package in R.

## Western blot analysis

For protein extraction, 5 mL of synchronous meiotic culture from time points indicated. Cells were collected and resuspended in 5 mL of 5% trichloroacetic acid and incubated on ice for 10 min. Cells were collected by centrifugation, supernatant was poured off, and protein was transferred to an Eppendorf tube with 500 μL of 1 M non-pHed Tris. Protein was collected by centrifugation, supernatant was poured off, and protein was resuspended in Tris-EDTA/250 mM DTT. SDS loading dye was added prior to boiling samples for 10 min. In all, 5 μL of protein sample was loaded onto a 4–15% Mini-PROTEAN TGX Precast polyacrylamide gel from Bio-Rad (4561086). Protein samples were resolved and transferred to a PVDF membrane using the Trans-Blot STurbo Transfer System. Membranes were blocked with 5% milk/TBST solution and then hybridized to antibodies indicated (Appendix Table S6). Anti-Rabbit-HRP from Kindle Biosciences (R1005) from Kindle Biosciences was used with 1-shot Digital-ECL substrate (R1003) to visualize protein. Images were taken with a digital camera.

## 1D recombination analysis

Samples were collected at the time points indicated, and cells killed with 0.1% sodium azide. For pulsed-field gel electrophoresis, cells were pelleted and embedded into low melting point agarose plugs. Genomic DNA was prepared within the plugs using zymolyase and proteinase K (Roche) and resolved in a 1% LE agarose gel in 0.5× TBE using a CHEF-DRII instrument (Bio-Rad) using the following settings: ChrVIII: 5 s–45 s ramp, 5.4 V/cm, 32 h, 14 °C; whole genome: 60 s switch time for 15 h followed by 90 s switch time for 9 h, 6 V/cm, 14 °C. For the analysis of specific hotspots, DNA was extracted from 10-ml meiotic culture and digested with XhoI (*HIS4-LEU2*) or HindIII (*GAT1*). Fragments were separated in a 0.8% agarose gel (Seakem LE agarose/1× TBE). Gels were then washed in ethidium bromide solution to visualize DNA fragments and imaged on a Biorad Gel Doc.

For Southern blot analysis, gels were transferred to a Hybond-XL nylon membrane (GE Healthcare) using alkaline capillary transfer. Probes (available in Appendix Table S5) were labeled with $^{32}$P-dCTP using the Prime-It RT random labeling kit (Agilent, catalog #300329) and hybridized to the membrane. The membrane was then exposed using a Fuji imaging screen and phospho-signal was detected on Typhoon FLA 9000 (GE). Signal quantifications were performed using ImageJ software. All experiments were performed at least twice. Graphs were constructed using Microsoft Excel.

## 2D recombination analysis

In total, 10 ml samples were collected at time points indicated and cross-linked with psoralen (0.1 mg/ml) and UV light to preserve secondary structures. Following the extraction of genomic DNA, samples were digested with XhoI. Samples were separated in the first dimension in a 0.4% gel in 1× TBE and run in 2.5 L 1× TBE at 40 V for 18 h. The gel was equilibrated in 1xTBE containing 0.5 μg/ml ethidium bromide. Lanes were cut out, positioned perpendicular, embedded in 0.8% agarose and run in 1× TBE containing 0.5 μg/ml ethidium bromide at 90 V for 15 h. DNAs was then transferred to a nylon membrane for Southern blot analysis as described for the 1D recombination analyses.

## Evolutionary analysis of meiotic HORMAD proteins

To generate HMM profiles for the Hop1 HORMA domain and CBR (PHD, wHTH, and HTH-C domains), we used hmmbuild from the hmmer package (version HMMER 3.1b1) (Eddy, 2011) based on multiple sequence alignments (MAFFT, v.7.149b (Katoh and Standley, 2013) "einsi" or "linsi") of Saccharomycetaceae-specific homologs. Two strategies were used to identify highly divergent domains: (1) Saccharomycetaceae domain-specific HMM profiles were used to query a dataset of new and previously identified HORMADs containing 149 proteins identified among 158 diverse eukaryotic genomes (Tromer et al, 2021; van Hooff et al, 2017) (Table EV1). (2) For multiple different seed sequences, with the N-terminal HORMA domain removed, we ran iterative searches using jackhmmer without any heuristic filters (option '--max') against 149 HORMAD homologs. Resulting alignments were manually inspected for the potential presence of divergent domains (especially for wHTHs). Domain identification from HMM profiles was confirmed by visual inspection of AlphaFold 2 and/or Omegafold 3D predicted structures for each identified HORMAD protein (Jumper et al, 2021; preprint: Wu et al, 2022). All AlphaFold 2 predicted 3D structures shown in Fig. 9 were downloaded from the AlphaFold Protein Structure Database (https://www.alphafold.ebi.ac.uk) (Varadi et al, 2022) version November 1, 2022 (AF-[Uniprot ID]-F1_model_v4.pdb files) or generated with AlphaFold Monomer version 2.0 or Omegafold using ColabFold (Mirdita et al, 2022).

# Data availability

The datasets and computer code produced in this study are available in the following databases: (1) Chip-Seq data and CC-Seq data: Gene Expression Omnibus GSE225129. Modeling computer scripts: GitHub https://github.com/hochwagenlab/Hop1-Loop2. (2) X-ray crystal structure of *V. polyspora* Hop CBR: Raw diffraction data is submitted to SBGrid Data Bank with entry ID 826, and final refined coordinates and structure factors are at the PDB with entry ID 7UBA. (3) Cryo-EM of *S. cerevisiae* Hop1 CBR plus nucleosome: Raw data is not submitted to EMPIAR as that database is intended for datasets that would have broad interest for tech development; final maps and coordinates are at the EMDB with entry ID 27030 (associated PDB entry 8CWW). Along with that, we also have a nucleosome-alone structure with EMDB entry ID 27096 (associated PDB entry 8CZE). Sequence Alignments: GitHub https://github.com/hochwagenlab/Hop1-Loop2.

# Peer review information

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

## Acknowledgements

We thank N Hollingsworth and A MacQueen for generously sharing antibodies. CM acknowledges support from the National Institutes of Health Ruth L. Kirschstein Postdoctoral Individual National Research Service Award (F32 GM139386). SU acknowledges support from the UCSD Molecular Biophysics Training Grant (National Institutes of Health T32 GM008326) and a National Science Foundation Graduate Research Fellowship program. ET acknowledges support from the Nederlandse Organisatie voor Wetenschappelijk Onderzoek (VI.Veni.202.223). AH acknowledges support from the National Institutes of Health (R01 GM111715 and R35 GM148223). KDC acknowledges support from the National Institutes of Health (R35 GM144121). We thank Olivia Micci-Smith, Nicole Adamski, Carolina Thornton, and everyone else at the NYU Center for Genomics and Systems Biology Genomics Core for sequencing experiments, as well as help troubleshooting the flow cytometry. This work was supported in part through the NYU IT High-Performance Computing resources, services and staff expertise. The authors acknowledge the facilities of the Cryo-EM Facility at UC San Diego, and technical assistance of R. Ashley on cryo-EM sample preparation and data collection. This work was partially conducted at the Advanced Light Source (ALS), a national user facility operated by Lawrence Berkeley National Laboratory on behalf of the U.S. Department of Energy, Office of Basic Energy Sciences, through the Integrated Diffraction Analysis Technologies (IDAT) program, supported by U.S. Department of Energy Office of Biological and Environmental Research. Additional support comes from the National Institutes of Health project ALS-ENABLE (P30 GM124169) and a High-End Instrumentation Grant S10OD018483. RMA, GGBB and MJN are supported by Wellcome Trust Investigator (200843/Z/16/Z) and Discovery (225852/Z/22/Z) Awards.

## Author contributions

**Carolyn R Milano**: Conceptualization; Resources; Data curation; Formal analysis; Supervision; Funding acquisition; Validation; Investigation; Visualization; Methodology; Writing—original draft; Project administration; Writing—review and editing. **Sarah N Ur**: Conceptualization; Data curation; Formal analysis; Investigation. **Yajie Gu**: Conceptualization; Data curation; Formal analysis; Validation; Investigation; Visualization. **Jessie Zhang**: Data curation; Formal analysis; Writing—review and editing. **Rachal Allison**: Data curation. **George Brown**: Data curation; Formal analysis; Validation. **Matthew J Neale**: Data curation; Formal analysis; Supervision; Validation; Visualization; Methodology; Writing—original draft; Writing—review and editing. **Eelco C Tromer**: Conceptualization; Data curation; Formal analysis; Validation; Visualization; Methodology; Writing—original draft; Writing—review and editing. **Kevin D Corbett**: Conceptualization; Resources; Data curation; Formal analysis; Supervision; Funding acquisition; Validation; Investigation; Visualization; Methodology; Writing—original draft; Project administration; Writing—review and editing. **Andreas Hochwagen**: Conceptualization; Resources; Data curation; Formal analysis; Supervision; Funding acquisition; Validation; Investigation; Visualization; Methodology; Writing—original draft; Project administration; Writing—review and editing.

## Disclosure and competing interests statement

The authors declare no competing interests.

# Expanded View Figures

**Figure EV1. Structure of a nucleosome + 2 Hop1 CBR complex.**

(A) Three views of cryo-EM density for a nucleosome + 2 Hop1 CBR complex (3.15 Å overall resolution, gaussian-smoothed with a σ of 1.1 Å). DNA is colored gray, histones colored yellow (H2A), red (H2B), blue (H3), and green (H4), and two Hop1 CBR domains colored yellow and orange. (B) Molecular model for a nucleosome + 2 Hop1 CBR complex, colored as in panel (A), except Hop1 CBR is colored blue (PHD), green (wHTH), and orange (HTH-C). (C) Schematic of the Widom 601 DNA sequence used for nucleosome assembly, with binding sites for Hop1 CBR #1 (blue) and #2 (green) noted. SHL: superhelical locations, 0 at the nucleosome dyad axis to 7 at the DNA ends. (D) Structural overlay of the DNA bound to Hop1 CBR #1 (blue) and #2 (green). In sequence alignment at bottom, solid lines indicate identity and dotted lines indicate shared status as either pyrimidine or purine.

**A**

Hop1 CBR #1

Hop1 CBR #2

45°    45°

**B**

Hop1 CBR #1

Hop1 CBR #2

45°    45°

**C**

-7   -6   -5   -4   -3   -2   -1   SHL 0   1   2   Hop1 CBR #1   4   5   6   7

5'-TGGAGAATCCCGGTGCCGAGGCCGCTCAATTGGTCGTAGACAGCTCTAGCACCGCTTAAACGCACGTACGCGCTGTCCCCCGCGTTTTAACCGCCAAGGGGATTACTCCCTAGTCTCCAGGCACGTGTCAGATATATACATCCTGT-3'
3'-ACCTCTTAGGGCCACGGCTCGGCGAGTTAACCAGCATCTGTCGAGATCGTGGCGAATTTGCGTGCATGCGCGACAGGGGGCGCAAAATTGGCGGTTCCCCTAATGAGGGATCAGAGGTCCGTGCACAGTCTATATATGTAGGACA-5'

Hop1 CBR #2

**D**

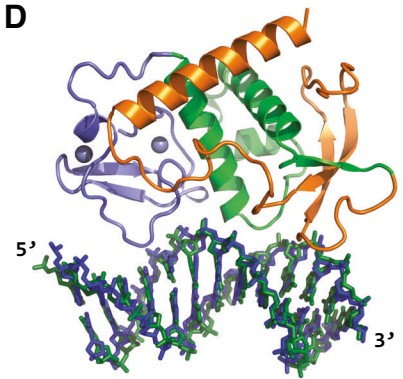

5'

3'

Site #1:  5'-AAGGGGATTACTC-3'
Site #2:  5'-CTCGGCACCGGGA-3'

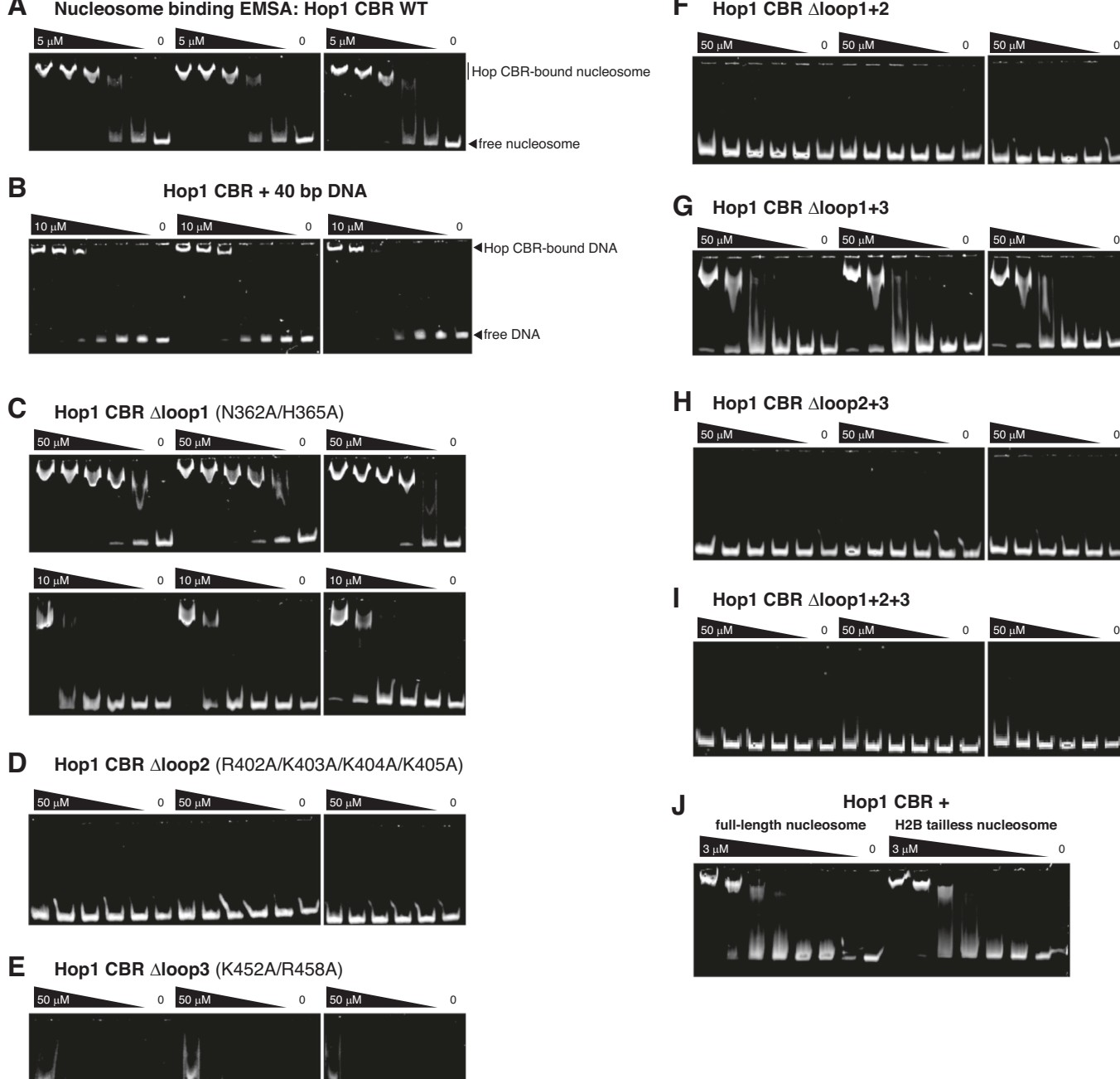

**Figure EV2. Nucleosome and DNA binding by the *S. cerevisiae* Hop1 CBR.**

(A) Triplicate electrophoretic mobility shift assays (EMSAs) for wild-type Hop1 CBR binding to reconstituted nucleosomes. The highest concentration of Hop1 CBR used was 5 μM, followed by serial 2× dilutions. Band intensities were quantified in ImageJ and $K_d$ was calculated in Prism using a single-site binding model ($K_d = 0.47 +/- 0.11$ μM). The representative gel image shown in Fig. 1F also appears here. (B) Triplicate EMSAs for wild-type Hop1 CBR binding a 40-bp DNA encompassing its preferred binding site on the Widom 601 DNA ($K_d = 2.1 +/- 0.6$ μM). (C) Triplicate EMSAs for Hop1 CBR Δloop 1 mutant (N362A/H365A) binding nucleosomes, with two different starting protein concentrations (top: 50 μM; bottom: 10 μM) ($K_d = 2.0 +/- 0.4$ μM). (D) Triplicate EMSAs for Hop1 CBR Δloop2 mutant (R402A/K403A/K404A/K405A) binding nucleosomes ($K_d > 50$ μM). The representative gel image shown in Fig. 1F also appears here. (E) Triplicate EMSAs for Hop1 CBR Δloop 3 mutant (K452A/R456A) binding nucleosomes ($K_d \sim 50$ μM). (F) Triplicate EMSAs for Hop1 CBR Δloop 1 + 2 mutant binding nucleosomes ($K_d > 50$ μM). (G) Triplicate EMSAs for Hop1 CBR Δloop 1 + 3 mutant binding nucleosomes ($K_d = 13 +/- 3$ μM). (H) Triplicate EMSAs for Hop1 CBR Δloop2 + 3 mutant binding nucleosomes ($K_d > 50$ μM). (I) Triplicate EMSAs for Hop1 CBR Δloop 1 + 2 + 3 mutant binding nucleosomes ($K_d > 50$ μM). The representative gel image shown in Fig. 1F also appears here. (J) EMSAs comparing binding of wild-type Hop1 CBR to nucleosomes reconstituted with full-length (left) or tailless (right) histone H2B.

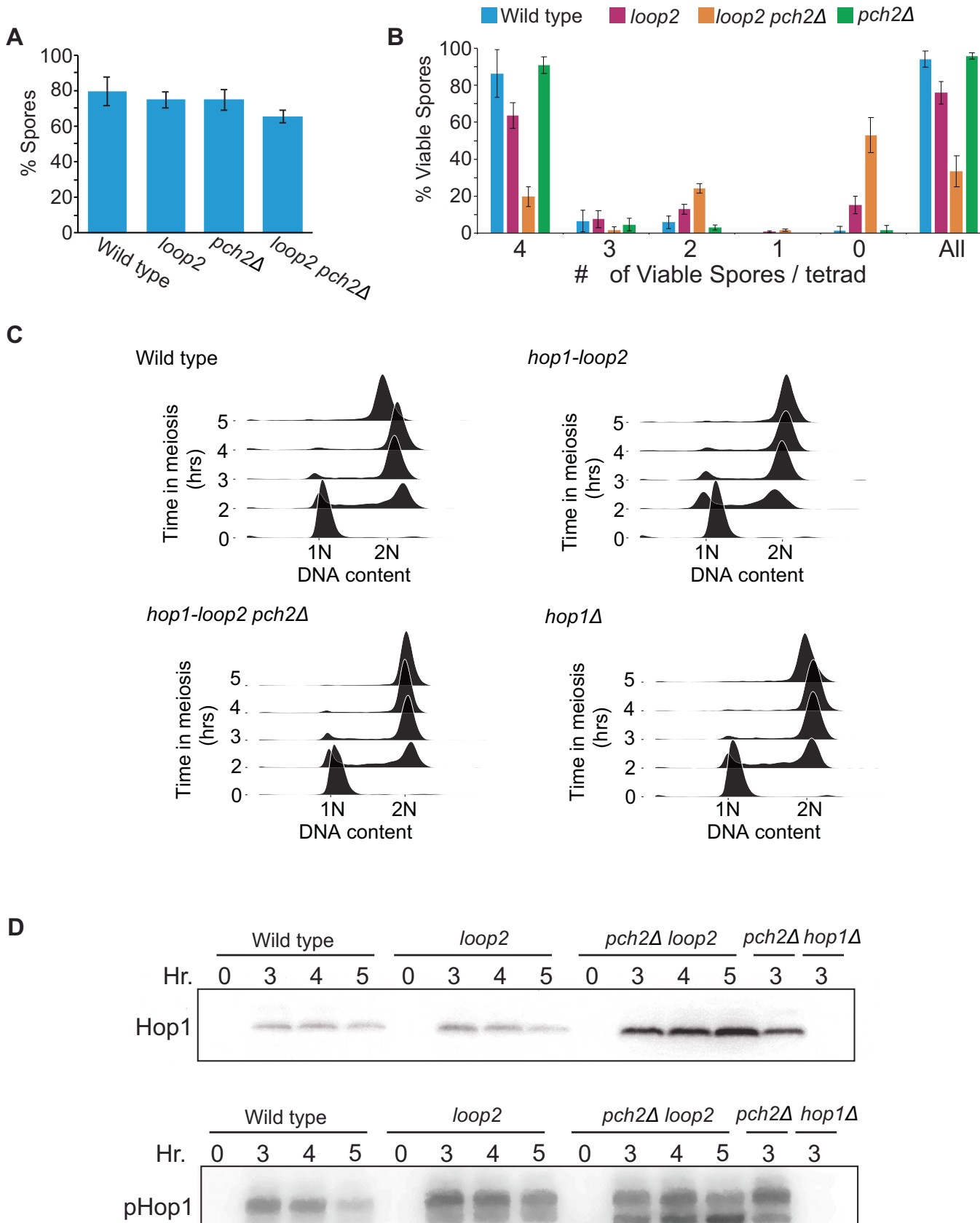

**Figure EV3. Hop1 protein and phosphorylation levels across genotypes.**

The *hop1-loop2* allele is simplified to *loop2* in figure labels. (A) Sporulation efficiency data for the indicated genotypes. This data represents at least four biological replicates. (B) Number of viable spores per tetrad for each genotype. At least 50 tetrads were dissected from each genotype. (C) Flow cytometry analysis of DNA content, monitoring DNA synthesis to indicate synchronous meiotic entry. This was performed every time samples were prepared from a meiotic yeast culture. (D) Western blots showing protein levels across genotypes for Hop1 protein (Top) (Subramanian et al, 2016), and for phosphorylated Hop1 T318 (Subramanian et al, 2016). This result was observed with at least three biological replicates. Source data are available online for this figure.

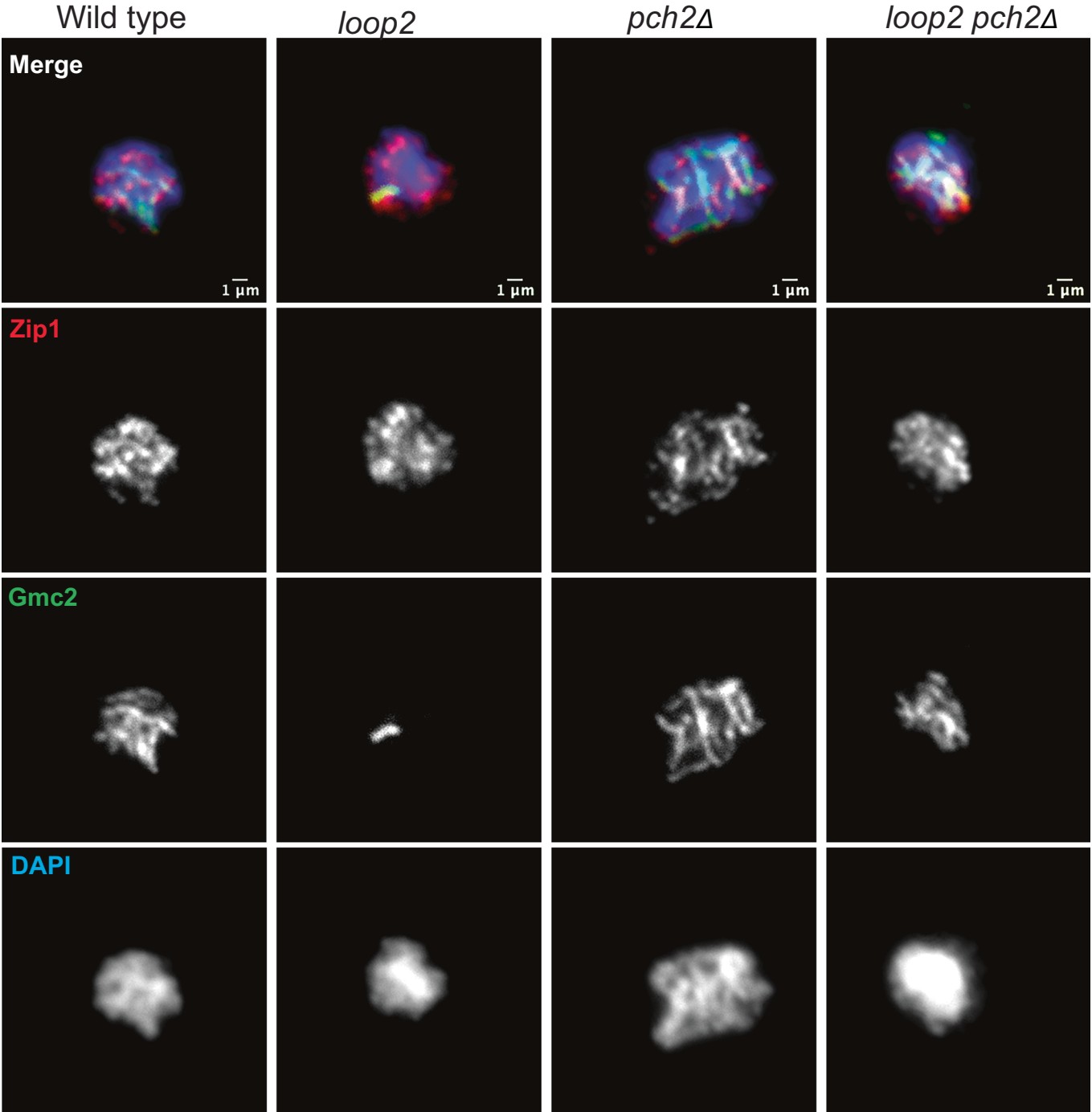

**Figure EV4. Localization of the central-element protein Gmc2.**

The *hop1-loop2* allele is simplified to *loop2* in figure labels. Samples were taken from meiotic cultures of the indicated genotypes at hour 3, chromosome spreads were prepared, hybridized with antibodies against Zip1 (pink) and Gmc2 (green), and stained with DAPI (blue). This was generated from samples taken from one meiotic time course. Source data are available online for this figure.

**wHTH:** Eukarya

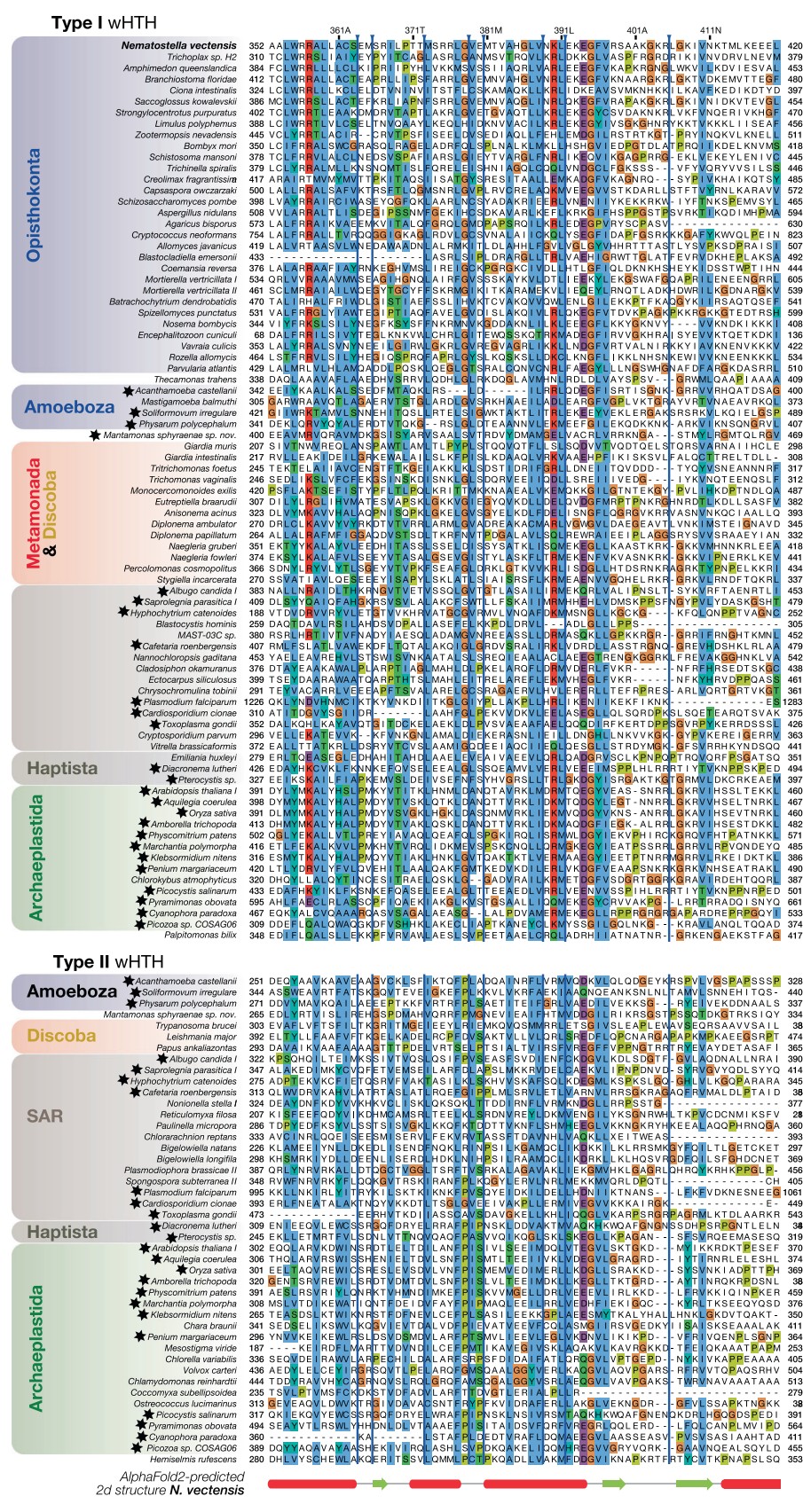

species with double wHTH topology

◀ **Figure EV5. Meiotic HORMAD wHTH domains can be classified into two types of which one of each type is present in meiotic HORMADs with a tandem wHTH configuration.**

Sequence alignment of wHTH domains found in 75 single wHTH and 25 double wHTH among 105 different eukaryotic species (Table EV1)—excluding those of the Saccharomycetaceae. To make the alignment more easily readable, we used the wHTH domain of *Nematostella vectensis* (Uniprot ID A7RLI6) as a reference and excluded any column in the multiple alignment that was not found in the sequence of this domain (see blue lines for marks of column removal). Full multiple alignments can be found in File S1 on https://github.com/hochwagenlab/Hop1-loop2. wHTH domains were manually classified into two types that correspond to either one of the wHTH domains found in meiotic HORMADs with two wHTH domains (see star). Phylogenetic tree analyses did not yield statistically sensible trees, likely due to the rather high sequence divergence found among the wHTH domains in these meiotic HORMAD proteins. Type 1 wHTHs are characterized by a conserved positively charged patch in the first alpha helix, while Type 2 wHTHs harbor a conserved 'FP' motif in a loop between two central alpha helices (see for classification Table EV1). Note that the wHTH domain is conserved until the first beta strand of the 'wing'. Most wHTHs in other meiotic HORMADs found among eukaryotes do have a second strand and a capping helix, but this part of the domain is highly divergent between lineages (i.e., where loop 3 in Saccharomycetaceae resides).

