## [Peer Review File · The EMBO Journal]

Chromatin binding by HORMAD proteins regulates meiotic recombination initiation

Carolyn Milano, Sarah Ur, Yajie Gu, Jessie Zhang, Rachal Allison, George Brown, Matthew Neale, Eelco Tromer, Kevin Corbett, and Andreas Hochwagen

Corresponding author(s): Andreas Hochwagen (andi@nyu.edu) , Kevin Corbett (kcorbett@ucsd.edu)

Review Timeline:

Submission Date:	10th May 23
Editorial Decision:	21st Jun 23
Revision Received:	1st Dec 23
Editorial Decision:	2nd Jan 24
Revision Received:	8th Jan 24
Accepted:	11th Jan 24

Editor: Hartmut Vodermaier

Transaction Report:

Dr. Andreas Hochwagen
New York University
Department of Biology
New York 10003

21st Jun 2023

Re: EMBOJ-2023-114485
Chromatin binding by HORMAD proteins regulates meiotic recombination initiation

Dear Andreas and Kevin,

Thank you for submitting your manuscript on Hop1 chromatin binding to The EMBO Journal, and apologies that it has taken a bit longer than usually to get back to you with a post-review decision. We have now received input from four referees with expertise in structural as well as cell biology and genetics of meiotic recombination, copied below for your information. As you will see, all referees consider your findings potentially interesting and valuable for the field, as well as generally well conducted. We would therefore be happy to pursue a revised version further for publication, pending adequate addressing of various experimental, conceptual, and presentational points raised in the four reports.

Since it is our policy to consider only a single round of major revision and therefore important to satisfactorily respond to all comments at the time of resubmission, I would invite you to consider the referees' comments together with your coworkers, and to prepare a tentative response letter detailing how each of the raised criticisms/queries might be answered/clarified. On the basis of this response, we could then discuss the requirements for a successful revision already during the early stages of the revision, e.g. via email or a follow-up video call. I should add that we could also offer extension of the default three-months revision period if needed, with our 'scooping protection' (meaning that competing work appearing elsewhere in the meantime will not affect our considerations of your study) remaining of course valid also throughout this extension.

Detailed information on preparing, formatting and uploading a revised manuscript can be found below and in our Guide to Authors. Thank you again for the opportunity to consider this work for The EMBO Journal, and I look forward to hearing from you in due time.

With best regards,

Hartmut

9) Digital image enhancement is acceptable practice, as long as it accurately represents the original data and conforms to community standards. If a figure has been subjected to significant electronic manipulation, this must be clearly noted in the figure legend and/or the 'Materials and Methods' section. The editors reserve the right to request original versions of figures and the original images that were used to assemble the figure. Finally, we generally encourage uploading of numerical as well as gel/blot image source data; for details see: embopress.org/page/journal/14602075/authorguide#sourcedata

At EMBO Press, we ask authors to provide source data for the main manuscript figures. Our source data coordinator will contact you to discuss which figure panels we would need source data for and will also provide you with helpful tips on how to upload and organize the files.

In the interest of ensuring the conceptual advance provided by the work, we recommend submitting a revision within 3 months (19th Sep 2023). Please discuss the revision progress ahead of this time with the editor if you require more time to complete the revisions. Use the link below to submit your revision:

Link Not Available

Referee #1:

In this manuscript, Milano et al use cryo-electron microscopy to report the structural basis of nucleosome DNA binding by Hop1, a budding yeast protein that is involved in chromosome axis formation in meiosis. They report that the PHD domain of Hop1's CBR binds to nucleosome DNA in a non-canonical manner, through the recognition of the bent DNA conformation rather than by binding to the histone tail. They identify a loop deletion that abrogates nucleosome binding, and use this in genetics, chip-seq and imaging experiments to show that Hop1 nucleosome binding plays a role in localisation to nucleosome-rich islands, and is required for normal formation of double-strand breaks and synapsis. They demonstrate that the CBR is found in the meiotic HORMAD proteins in many organisms across the phylogenetic tree, although notably not in mammals.

My assessment of this manuscript is based largely in my expertise in protein structure, and other reviewers must assess the technical details of the yeast genetics and chip-seq data. The structural data are very convincing. The choice of a different yeast species for crystallography appears appropriate given the changes in crystallising the *S. cerevisiae* Hop1-CBr, and the structure solution is clearly highly reliable. Similarly, the cryo-EM data are convincing and provide clear validation that the CBR structure is conserved between *V. polyspora* and *S. cerevisiae*.

Overall, the manuscript reports the structural basis of an important interaction between Hop1 and nucleosomes, and the functional consequence of its disruption. This work is of clear interest to the meiosis field, especially those studying yeast meiosis in which this is directly applicable. The data are well presented and support the conclusions made. The manuscript is certainly suitable for publication subject to a couple of very minor revisions that I have outlined below.

Minor comments

How was the crystal structure solved? The authors state SAD but don't explain this further. Given the X-ray wavelength and number of sites, I assume they used the SAD signal from the zinc absorption K-edge. This should be stated explicitly.

The authors should show sample electron density from the structure - preferably a 2Fo-Fc map and anomalous difference map around the two zinc sites.

The text states "The isolated *S. cerevisiae* Hop1 CBR bound nucleosomes with a Kd of 760 nM, compared to 2.1 μ M for DNA alone", and "... reducing the Kd from 760 nM to 2.0 μ M", whereas figures 1F,G and S7 give values of 0.5 and 2.1 μ M. This disparity should be corrected.

The deletion data for the three loops nicely demonstrates their roles in nucleosome binding. It would be nice to see the same experiments performed with free DNA to see whether the loops have differential roles in binding to nucleosomal or free DNA.

Referee #2:

General

This study explores the mechanism by which the key HORMAD-domain-containing protein Hop1 binds to nucleosomes during meiotic prophase in budding yeast. HORMADs are evolutionarily conserved and critical for meiotic recombination regulation (both Spo11-DSB formation and repair), so elucidating this mode of action is important and of general interest to the recombination community.

Overall, the data presented are novel, high quality, and convincing. However, in many places-in part due to the extent of data that are covered-the reader is frequently left without clear description of how the observations support the conclusions stated in the text. Many figures and panels have only a cursory mention in the text. In other places, key conclusions lack rigorously experimental and/or quantitative support (for example the impact on Hop1 binding and DSB formation in "island" regions.). I am sure that the authors can easily sort out these problems and make the study both easier to read and more quantitatively convincing.

Major concern:

My major concern-that is not easily addressed-is how the authors can infer that these phenotypes of the Hop1 loop mutants are specific to the loss of histone binding? Would one not expect the same results from any Hop1 hypomorph? How might this concern be dealt with?

Specific comments:

MINOR: Fig 1A. Is this *S. cerevisiae*? Could an equivalent line diagram of the *V. polyspora* be added for comparison underneath here? Could amino acid numbers for the residues start/stop of each named domain be added? This info would all help to better link the figure to the text describing it.

MINOR: Fig 2C. Could the Hop1 ChIP be expressed also as ratios to better demonstrate the proposed loss of binding in the Hop1-loop2 etc mutants? It is very hard to tell what is going on from these whole-chromosome plots. Perhaps some additional partial zoom ins to chromosomal subdomain scale would help?

MINOR: Page 5. "The overall structure of the nucleosome-bound *S. cerevisiae* Hop1 CBR closely matches our crystal structure of the *V. polyspora* Hop1 CBR (Figure S5)". Can this statement be explained more carefully? The *S. cerevisiae* structure (I presume) is based on an inference from the cryoEM? The sentence glosses over this difference. Was the Sc "structure" modelled upon the Vp crystal structure, thus creating a circular argument about similarity? The main point is to be clear in the text that there are two types of structures being compared-one crystallographic and one from cryoEM.

MAJOR: General - frequently data are presented (often in many Supplementary figures) but given little or no description in the main text. For example in Fig S7, no mention of binding (or not) to the 40 bp dsDNA is described (panel J), despite this being a major point introduced in the opening sentence of this section (page 6 paragraph 2). Adding direct referencing to each panel (and keeping the panel description in the order they are referred to in the text) would help greatly to see where the observations are located that support the author's conclusions. (For example Fig S8 is also referenced as a whole, never individually).

MAJOR: Page 6. Fig 2BC. "With a specific loss of Hop1 enrichment from island regions". Where is this demonstrated? It is very hard to see this effect in Fig 2C. The rest of this paragraph also makes statements concerning altered binding (or not) without any figures or quantitative support. A specific effect in island regions is unconvincing (as currently presented). Revision of this text is essential. Perhaps the authors meant to refer to panel 2D? If so, how were these stated fold changes quantified from these aggregate data? Or, maybe, as suggested above, some representative zoom ins could help?

Secondly, given the anticipated differential effects at different loci (based on the Hop1 results), the DSB data (Fig 4) would be

more convincing (i.e. to directly link DSB frequency observations to the reported differential Hop1 binding) if a genome-wide analysis of DSB formation were performed.

MINOR: page 7 and Fig 2E. Meta analysis is prone to skew (For example, perhaps the effect is driven by a change at only one of the 16 centromeres.) To be convincing, it would be preferable to show some non-aggregated data to demonstrate the increased Hop1 data around CENs. Is this effect visible in panel C for example? Where are the CENs?

If only Rec8-dependent recruitment of Hop1 was retained, wouldn't the map look very different? How do the effects of the loop mutant compare spatially to the effect of Red1 binding in Hop1 delta or PHD mutants the authors previously published?

MINOR: The discussion regarding two potential conformations of Hop1 (state 1 and state 2) is interesting-albeit highly speculative. Are there alternative models that the data also support?

Referee #3:

The manuscript by Milano et al is focused on the "chromatin binding region" of the budding yeast Hop1 protein that contains PHD, wHTH and HTH-C subdomains. They first crystallized the CBR from *V. polyspora* and then obtained a high resolution cryo-EM structure of the *S. cerevisiae* CBR bound to a nucleosome. Mutation of amino acids contained in loop2 from the wHTH domain, disrupted nucleosome binding in gel mobility shift assays. The authors then phenotypically characterized the hop1-loop2 mutant in meiosis. They conclude that nucleosome binding facilitates Hop1 binding to small chromosomes and "islands" previously defined by the Hochwagen lab, DSB formation in some regions of the genome, as well as chromosome synapsis and spore viability. Hop1 is a member of a conserved family of axis proteins and plays a critical role in meiotic recombination. The authors performed an interesting phylogenetic analysis of the CBR from different species and found that, while not all of the subdomains are conserved, the ability to bind chromatin is likely conserved. This is an interesting paper, although the modest effects on DSB formation and spore viability suggest that nucleosome binding is not essential for Hop1 function in meiosis.

Major comments:

1. One thing that is curious is that the Hop1 CBR specifically binds between SHL2.5 and 3.5 position on the nucleosome. The authors rule out DNA sequence and histone tail binding as the basis for this specificity and instead cite "bent DNA". Is there any evidence that the bend in the DNA at this position is different from that at other places around the histone core? If not, how does the bent DNA confer specificity?
2. For the data in Figure 2A, does the distribution of viable spores in tetrads suggest that the decrease in spore viability is due to Meiosis I non-disjunction? This would be expected if decreased Hop1 binding resulted in increased IS recombination and therefore fewer crossovers. While an increase in IS JMs was not observed at the HIS4-LEU2 hotspot (although the JM data are not compelling), this single hotspot may not be indicative of the genome as a whole. If the spore viability pattern does not support MI non-disjunction as the reason for the dead spores, what ideas do the authors have for the basis of the spore inviability?
3. In Figure 3B, the authors assert that there is a "defect in chromosomal association of Hop1" since at the 2 hour timepoint the Hop1-loop2 protein covers only 20% of the nuclear area occupied by Hop1. However the values for both proteins at the 3 hour timepoint appear very similar and are perhaps elevated at 4 hours for Hop1-loop2. Therefore if there is a defect in chromosomal association it is overcome pretty rapidly. This conclusion should be weakened to reflect the data.
4. In Figure 4E put arrows on the insets indicating the spots corresponding to the IH and IS JMs. The signal for hop1-loop2 is much fainter than WT and I don't see the IS JMs. The graph suggests that the JMs formed in hop1-loop2 are similar to hop1-loop2 pch2 but the JM spots for hop1-loop2 pch2 are not convincing. How many times was this experiment done?

Minor comments:

Page 3: rather than "...Mek1, which enforces DSB repair via the homologous chromosome...", say "...Mek1, which biases DSB repair towards using the homologous chromosome..."

, bottom: "...higher average rates of recombination..." should be higher average frequencies of recombination..." as a rate requires a time component (eg miles/hour)

In the references, gene and genus species names should be italicized (eg. 38). Only the first word and proper nouns should be italicized (eg, 28).

Page 6, The text says that the ScHop1 CBR binds to nucleosomes with a Kd of 760 nM, but the table in Figure 1G say 0.5 μ M.

Shouldn't this be 0.76 μ M?

Page 7, third paragraph, In wild-type cells...

In the fourth paragraph, wild-type Hop1..

Fourth paragraph, needs a citation for the statement that Zip1 is a transverse filament protein that is a marker for chromosome synapsis.

The heading on page 8 should be *hop1-loop2* (italicized) spore survival and DSB formation depend (not depends) on *PCH2* (italicized). The experiment was genetic, not biochemical.

Page 9 first paragraph wild-type localization...

Figure 1 legend: For F, it should be indicated that the nucleosomes contain histones derived from humans

Figure 2A: Why are significance values not given for WT vs. *loop2 tel* and other combinations? Spore viability is highly quantitative and my guess from looking at the bar graphs is that the rescue of *hop1-loop2* by *tel1* is not complete (I agree with the authors that it is to "near WT levels"). The p-values should be shown to confirm this.

It would be helpful if the green dots indicating centromeres in Figure 2C were more prominent.

In Figure 4C, why do some of the time points lack error bars? Were these timepoints only measured in one replicate? Similarly there are no error bars for the curves in 4D. How many times was this experiment repeated?

The authors found that the spore viability of *hop1-loop2* is partially rescued by *tel1*, which they infer is because of an increase in DSB formation. It may be that the relatively high spore viability of the *hop1-loop2* alone is also due to increased DSBs occurring due to the defect in synapsis, since synapsis is necessary for downregulating Spo11 activity (eg. Murakami, Keeney *ecm11* paper).

Table S5: gene names should be italicized, the "a" in *MATa* should be in bold,

H8644, *LEU2/leu2 Δ 0*. For the references, use the same numbering as the Bibliography. If blanks indicate strains that are unpublished, use "this work".

kanMX, not *KanMX*

H11569, remove *LYS2/LYS2*-genotypes for homozygous wild-type alleles are not included

Figure S8. Have the p318-Hop1 antibodies been previously characterized and validated? If so, please supply a reference, if not, please show validation here. Why does the pHop1 antibody recognize a doublet in the *hop1-loop2*, *pch2*, and *pch2 loop2* diploids? Only a single species is observed with the total Hop1 antibody.

Referee #4:

The conserved meiotic axis protein protein Hop1 (and its homologs in other species) lies at the center of meiotic chromosome dynamics and recombination. As the penultimate component of the meiotic chromosome axis, Hop1 has critical roles in the initiation of meiotic recombination by double strand break formation, in the direction of meiotic DSB repair away from the sister chromatid and towards the homolog, and in modulating homolog pairing and synapsis. Most previous studies of Hop1 have focused on the role of Hop1's HORMA domain in recruiting Hop1 to the meiotic axis, via interaction with the meiotic axis protein Red1 (and analogs in other species), but Hop1 also has been shown to have DNA binding activity, suggesting that it is also recruited directly to chromatin. In the current work, Milano et al. combined structural biology, biochemistry, high-throughput ChIP-seq analysis, meiotic cytology and molecular analysis of meiotic recombination to identify roles for this second activity, residing in the C-terminal half of the protein, in Hop1 function. This is an important work that has the potential to substantially advance understanding in all organisms that undergo meiosis, and the data are compelling. I do have some disagreements regarding interpretation, but these most likely can be addressed by text revision, some additional analysis of the existing data, and possibly a few additional experiments that could readily be accomplished.

Major comments:

1. Hop1 binding to nucleosomes/DNA.

a. Previous work (predominantly by Muniyappa's group) has examined Hop1 DNA binding activity, but is not cited anywhere in the current manuscript. While Muniyappa's penultimate conclusion, that Hop1's physiological substrate is G-quadruplex, is most likely incorrect, there are still useful data in these papers and they ought to be at least briefly mentioned and discussed.

b. With regards to binding to nucleosomes vs. DNA, one notes that the Widom sequence used for structural and EMSA studies is quite G/C rich, and the question arises as to whether or not some kind of sequence content preferences are impacting where Hop1 is observed to bind, and the fact that under conditions of Hop1 excess the predominant species still only contains one Hop1 bound. While the Widom sequence is likely critical for getting a complex that is tractable for structural studies, it should not be necessary for EMSA studies. Authors do discuss this, but it would be useful to test the role of the specific target sequence in the measured affinities. Possible tests could include mutating the primary bound sequence to match the secondary bound sequence, shuffling sequences (while retaining G/C content), etc-up to the authors to decide. At the very least, Hop1 binding to a 40 nt oligo containing the secondary bound sequence needs to be done. It would also seem important, for the sake of completeness, to examine binding to DNA alone by the loop 2 mutant that abolishes nucleosome binding. It may turn out that previous studies of Hop1 binding to DNA will supply insight into appropriate substrates.

This may seem nit-picky and asking for a grab-bag of additional work, but given the argument that Hop1 preferentially binds to nucleosome-rich regions, it seems important to know whether the primary determinant of Hop1 binding to these regions is the density of nucleosomes per se, or underlying sequence characteristics that attract both Hop1 and nucleosomes independently. If this is not done, then the text presenting and discussing the data throughout the manuscript needs to be substantially revised to give more weight to possible sequence preference.

In addition, because *in vitro* binding experiments were done with the CBR rather than full-length Hop1, the question arises whether or not there are additional domains of Hop1 that contribute to chromatin binding. Addressing this may be pushing the bounds of the scope of the paper, but if it is not addressed then the appropriate caveats should be included.

2. ChIP-Seq.

a. From a cursory examination of the data in Figure 2C, it appears that the axis islands are not the only regions that are affected by loop 2 mutation. Because these are calibrated ChIPs, it should be possible to directly subtract the loop 2 mutant signal from wild type (similarly for other combinations), to identify regions that are differentially affected.

b. Figures 2D and 2E would be more readily interpreted if the Y-axis scales were the same in each panel-i.e. 1.0 to 3.5 in all panels in 2D, etc. It appears that the effect of loop 2 mutants is both to decrease axis island binding AND to increase non-island binding, in both PCH2 and pch2. If this is true, it would call for some re-interpretation. As a minor point, the colors used in these figure panels may not be color-blind friendly-something to check.

3. Chromosome spreads versus ChIP-Seq.

The differences, in terms of impact of loop 2 and pch2 mutation, between these two assays might be understood if ChIP-seq mostly measures the density of the Hop1 immediately adjacent to chromatin, while signal intensity on chromosome spreads, of course, detects every Hop1 monomer in the Hop1 chain. While this is considered briefly, it would be useful to discuss it more prominently, as many readers might not be aware of this important issue. For example, my impression is that the data might indicate that Pch2 is more active in remodeling/removing Hop1 in chains, and have less impact of chromatin-adjacent/bound Hop1.

Re: EMBOJ-2023-114485

“Chromatin binding by HORMAD proteins regulates meiotic recombination initiation”

Dear Referees,

Thank you all for your insightful and constructive feedback on our manuscript. In preparing the updated manuscript, we have taken special care to address every concern you've raised.

Below we have copied all of the Referees' responses. You will find our response to each raised concern in **Bold**.

Referee #1:

In this manuscript, Milano et al use cryo-electron microscopy to report the structural basis of nucleosome DNA binding by Hop1, a budding yeast protein that is involved in chromosome axis formation in meiosis. They report that the PHD domain of Hop1's CBR binds to nucleosome DNA in a non-canonical manner, through the recognition of the bent DNA conformation rather than by binding to the histone tail. They identify a loop deletion that abrogates nucleosome binding, and use this in genetics, chip-seq and imaging experiments to show that Hop1 nucleosome binding plays a role in localisation to nucleosome-rich islands, and is required for normal formation of double-strand breaks and synapsis. They demonstrate that the CBR is found in the meiotic HORMAD proteins in many organisms across the phylogenetic tree, although notably not in mammals.

My assessment of this manuscript is based largely in my expertise in protein structure, and other reviewers must assess the technical details of the yeast genetics and chip-seq data. The structural data are very convincing. The choice of a different yeast species for crystallography appears appropriate given the changes in crystallising the *S. cerevisiae* Hop1-CBr, and the structure solution is clearly highly reliable. Similarly, the cryo-EM data are convincing and provide clear validation that the CBR structure is conserved between *V. polyspora* and *S. cerevisiae*.

Overall, the manuscript reports the structural basis of an important interaction between Hop1 and nucleosomes, and the functional consequence of its disruption. This work is of clear interest to the meiosis field, especially those studying yeast meiosis in which this is directly applicable. The data are well presented and support the conclusions made. The manuscript is certainly suitable for publication subject to a couple of very minor revisions that I have outlined below.

Minor comments

How was the crystal structure solved? The authors state SAD but do not explain this further. Given the X-ray wavelength and number of sites, I assume they used the SAD signal from the zinc absorption K-edge. This should be stated explicitly.

We apologize for the ambiguity. Indeed, as the reviewer notes, the structure was determined using single-wavelength anomalous diffraction (SAD) phasing with the natively bound zinc ions. This is now explicitly noted in both the *Results* and *Materials and Methods* sections.

The authors should show sample electron density from the structure - preferably a 2Fo-Fc map and anomalous difference map around the two zinc sites.

We now show sample 2Fo-Fc electron density around one zinc coordination site and also show the anomalous difference map for the same area in the new Appendix Figure 2.

The text states "The isolated *S. cerevisiae* Hop1 CBR bound nucleosomes with a K_d of 760 nM, compared to 2.1 μ M for DNA alone", and "... reducing the K_d from 760 nM to 2.0 μ M", whereas figures 1F,G and S7 give values of 0.5 and 2.1 μ M. This disparity should be corrected.

We apologize for these errors, which were based on earlier measurements of binding affinity for the wild-type protein. These statements have been corrected so that the text is consistent with the figure.

The deletion data for the three loops nicely demonstrates their roles in nucleosome binding. It would be nice to see the same experiments performed with free DNA to see whether the loops have differential roles in binding to nucleosomal or free DNA.

This is an interesting question. We chose not to pursue these experiments because overall, our data show that the most biologically relevant interactions are between the Hop1 CBR and nucleosome-wrapped, bent DNA.

Referee #2:

General

This study explores the mechanism by which the key HORMAD-domain-containing protein Hop1 binds to nucleosomes during meiotic prophase in budding yeast. HORMADs are evolutionarily conserved and critical for meiotic recombination regulation (both Spo11-DSB formation and repair), so elucidating this mode of action is important and of general interest to the recombination community.

Overall, the data presented are novel, high quality, and convincing. However, in many places-in part due to the extent of data that are covered-the reader is frequently left without clear description of how the observations support the conclusions stated in the text. Many figures and panels have only a cursory mention in the text. In other places, key conclusions lack rigorously experimental and/or quantitative support (for example the impact on Hop1 binding and DSB formation in "island" regions.). I am sure that the authors can easily sort out these problems and make the study both easier to read and more quantitatively convincing.

Thank you for your feedback, we have substantially edited the manuscript to include more detailed descriptions of figures and individual figure panels, as appropriate. In addition, we substantially expanded the experimental and quantitative analysis of Hop1 binding in island regions (Figure 3) and around centromeres (Figure 4). The revised manuscript now also includes genome-wide analysis of DSBs (cc-seq, conducted by the Neale lab) and demonstrate that, consistent with the CBR-dependent changes of Hop1, nucleosome binding affects DSB activity in island regions and on short chromosomes (Figure 7).

Major concern:

My major concern-that is not easily addressed-is how the authors can infer that these phenotypes of the Hop1 loop mutants are specific to the loss of histone binding? Would one not expect the same results from any Hop1 hypomorph? How might this concern be dealt with?

The *hop1-loop2* mutant is unlikely to be a general hypomorph for two reasons. First, the protein levels of Hop1 are unchanged in *hop1-loop2* cells, indicating that the reduced function is not a simple issue of reduced Hop1 protein available (EV Figure 3D). Western analyses also show normal levels of Hop1 phosphorylation and normal increases of Hop1 levels in *pch2* mutants, suggesting that these aspects of Hop1 regulation are unaffected by the *hop1-loop2* mutation (EV Figure 3D). Second, although several functions of Hop1 are clearly defective in the *hop1-loop2* mutant, other functions, most notably the establishment of interhomolog bias is unaffected by the *hop1-loop2* mutation (Figure 8). Based on these data, we argue that the phenotypes of *hop1-loop2* represent a separation of functions, that specifically highlights roles of Hop1 that require histone binding. We now included these arguments in the discussion.

Specific comments:

MINOR: Fig 1A. Is this *S. cerevisiae*? Could an equivalent line diagram of the *V. polyspora* be added for comparison underneath here? Could amino acid numbers for the residues start/stop of each named domain be added? This info would all help to better link the figure to the text describing it.

We have updated Figure 1A to include residue numbers for the HORMA domain and CBR for both *S. cerevisiae* and *V. polyspora* Hop1

MINOR: Fig 2C. Could the Hop1 ChIP be expressed also as ratios to better demonstrate the proposed loss of binding in the Hop1-loop2 etc mutants? It is very hard to tell what is going on from these whole-chromosome plots. Perhaps some additional partial zoom ins to chromosomal subdomain scale would help?

We now depict the total ratio of Hop1 ChIP binding across genomes in Figure 2B. Now, the chromosome plots display Hop1 binding for each genotype discussed (normalized to wild type) overlaid with wild-type ChIP binding for easier comparison, and we also included partial zoom-in panels for chromosome subdomains (Figure 2D), highlighting a centromere region and island region. Taking advantage of the spike-in normalized ChIP data, we also directly compare the binding in wild type to the *loop2* mutants by subtracting the mutant from wild-type ChIP data and plotting the remaining signal (Figure 2E).

MINOR: Page 5. "The overall structure of the nucleosome-bound *S. cerevisiae* Hop1 CBR closely matches our crystal structure of the *V. polyspora* Hop1 CBR (Figure S5)". Can this statement be explained more carefully? The *S. cerevisiae* structure (I presume) is based on an inference from the cryoEM? The sentence glosses over this difference. Was the Sc "structure" modelled upon the Vp crystal structure, thus creating a circular argument about similarity? The main point is to be clear in the text that there are two types of structures being compared—one crystallographic and one from cryoEM.

We have slightly altered the sentence in question to read “The overall structure of the nucleosome-bound *S. cerevisiae* Hop1 CBR in our cryoEM map closely matches our crystal structure of the *V. polyspora* Hop1 CBR (overall Ca r.m.s.d. of 1.06 Å; New Appendix Figure S5).”

We feel that the meaning of this sentence is clear: the structure of the *S. cerevisiae* Hop1 CBR (as unambiguously determined by cryoEM to 2.74 Å resolution) is very similar to that of the *V. polyspora* Hop1 CBR (as unambiguously determined by X-ray crystallography to 1.55 Å resolution). The cryoEM map for the *S. cerevisiae* Hop1 CBR is easily interpretable/buildable, and the structure of the CBR was refined against the cryoEM map with the only added positional restraints being used to maintain the position and geometry of the zinc ions relative their coordinating amino acids. Since both structures were experimentally determined to better-than-3 Å resolution, there is no need to infer anything about the *S. cerevisiae* structure from the earlier *V. polyspora* structure.

MAJOR: General - frequently data are presented (often in many Supplementary figures) but given little or no description in the main text. For example in Fig S7, no mention of binding (or not) to the 40 bp dsDNA is described (panel J), despite

this being a major point introduced in the opening sentence of this section (page 6 paragraph 2). Adding direct referencing to each panel (and keeping the panel description in the order they are referred to in the text) would help greatly to see where the observations are located that support the author's conclusions. (For example Fig S8 is also referenced as a whole, never individually).

We have revised the text to ensure that all figures/panels are explicitly explained, referenced, and when appropriate, called upon in the order in which they appear.

MAJOR: Page 6. Fig 2BC. "With a specific loss of Hop1 enrichment from island regions". Where is this demonstrated? It is very hard to see this effect in Fig 2C. The rest of this paragraph also makes statements concerning altered binding (or not) without any figures or quantitative support. A specific effect in island regions is unconvincing (as currently presented). Revision of this text is essential. Perhaps the authors meant to refer to panel 2D? If so, how were these stated fold changes quantified from these aggregate data? Or, maybe, as suggested above, some representative zoom ins could help?

We now include partial zoom-in panels for chromosome subdomains, including a centromere region and island region in figure 2D. The differential Hop1 binding can be visualized particularly well in figure 2E (see Appendix Figure S7 for the whole genome). We have dedicated Figure 3 to our analysis of island binding, and this includes heatmaps of Hop1 binding in island regions and non-island regions in wild type vs *loop2*, and shows a pronounced decrease in Hop1 binding specifically from island regions(Figure 3A). Meta-plots in Figure 3A indicate this difference is statistically significant (non overlapping 95% confidence intervals). Together, these data show a clear effect on individual island regions to complement the averaged profiles comparing island and non-island regions.

Secondly, given the anticipated differential effects at different loci (based on the Hop1 results), the DSB data (Fig 4) would be more convincing (i.e. to directly link DSB frequency observations to the reported differential Hop1 binding) if a genome-wide analysis of DSB formation were performed.

We now include genome-wide DSB analysis as determined by cc-seq (Figure 7). These analyses show that Hop1 nucleosome binding affects DSB activity in island regions and on short chromosomes (Figure 7B-C), consistent with the CBR-dependent changes in Hop1 binding in these regions.

MINOR: page 7 and Fig 2E. Meta analysis is prone to skews (For example, perhaps the effect is driven by a change at only one of the 16 centromeres.) To be convincing, it would be preferable to show some non-aggregated data to demonstrate the increased Hop1 data around CENs. Is this effect visible in panel C for example? Where are the CENs?

The point of the reviewer is an important one. We have now complemented the meta-analyses of Hop1 binding with non-aggregated data, including heatmaps of all axis association sites in island and non-island regions (Figure 3A), zoom-ins of sample regions (Figure 2D), as well as traces of Hop1 binding around centromeres (Figure 4B). These data show that the observed effects are general and not driven by individual outliers.

The position of the centromere in Figure 2D, 2E and Appendix Figure S7 is now indicated by a triangle.

If only Rec8-dependent recruitment of Hop1 was retained, wouldn't the map look very different? How do the effects of the loop mutant compare spatially to the effect of Red1 binding in Hop1 delta or PHD mutants the authors previously published?

We previously published analyses for both Red1 and Hop1 (Sun et al., 2015; Heldrich, Milano, et al., 2022) and showed that Rec8-dependent recruitment is the major pathway for both Red1 and Hop1. This pathway is responsible for most of the focal enrichment peaks, which dominate the enrichment profile of wild-type cells, whereas the Hop1-PHD dependent recruitment manifests as broader regions of enrichment that further boost overall Hop1 and Red1 enrichment in the islands. In other words, Rec8-dependent recruitment is solely responsible for axis protein enrichment in non-island regions, whereas both Rec8-dependent and Hop1-PHD-dependent contribute additively to axis enrichment in islands. The binding profiles of the *hop1-loop2* mutant look indistinguishable from the previously published profiles of the *hop1-phd* mutants in that they show specific depletion of Hop1 from island regions. Like for the *hop1-phd* mutant, all axis binding is lost when *hop1-loop2* is combined with a *rec8* mutation, thus eliminating both recruitment pathways (Figure 3B). These data indicate that the Rec8-independent recruitment of axis proteins relies on histone binding by the CBR. We now specifically stress the similarity of these profiles in the text.

MINOR: The discussion regarding two potential conformations of Hop1 (state 1 and state 2) is interesting-albeit highly speculative. Are there alternative models that the data also support?

We have now amended the discussion to include several additional models that could be invoked and compare them against the available data.

Referee #3:

The manuscript by Milano et al is focused on the "chromatin binding region" of the budding yeast Hop1 protein that contains PHD, WTH and HTH-C subdomains. They first crystallized the CBR from *V. polyspora* and then obtained

a high resolution cryo-EM structure of the *S. cerevisiae* CBR bound to a nucleosome. Mutation of amino acids contained in loop2 from the wHTH domain, disrupted nucleosome binding in gel mobility shift assays. The authors then phenotypically characterized the hop1-loop2 mutant in meiosis. They conclude that nucleosome binding facilitates Hop1 binding to small chromosomes and "islands" previously defined by the Hochwagen lab, DSB formation in some regions of the genome, as well as chromosome synapsis and spore viability. Hop1 is a member of a conserved family of axis proteins and plays a critical role in meiotic recombination. The authors performed an interesting phylogenetic analysis of the CBR from different species and found that, while not all of the subdomains are conserved, the ability to bind chromatin is likely conserved. This is an interesting paper, although the modest effects on DSB formation and spore viability suggest that nucleosome binding is not essential for Hop1 function in meiosis.

Major comments:

1. One thing that is curious is that the Hop1 CBR specifically binds between SHL2.5 and 3.5 position on the nucleosome. The authors rule out DNA sequence and histone tail binding as the basis for this specificity and instead cite "bent DNA". Is there any evidence that the bend in the DNA at this position is different from that at other places around the histone core? If not, how does the bent DNA confer specificity?

We don't have a full explanation for why we observe semi-specific binding of the Hop1 CBR to the observed site at SHL 2.5 to 3.5. First, we would note that we do observe a secondary binding location at SHL -5.5 to -6.5. These two sites share no common sequence in the Widom 601 nucleosome positioning sequence we used for the structure. We observe that in both binding sites, the DNA minor groove is significantly opened as part of the DNA bending, and both loop 2 and loop 3 bind in this wide minor groove. As the reviewer notes, however, other sites around the nucleosome share the same bent DNA structure but do not appreciably bind Hop1. Our tentative explanation is that specificity is governed by a combination of DNA bending and sequence specificity. One additional factor to note is that, given how single-particle cryoEM data is processed, there could be minor populations of nucleosomes bound to Hop1 at different positions, but these would be averaged out and extremely difficult to detect without a much larger dataset.

Overall, while we do observe two semi-specific binding locations for the Hop1 CBR on a nucleosome wrapped with the Widom 601 sequence, in cells we expect that nucleosome binding is largely non-sequence specific, and that Hop1 could bind any of the bent DNA segments around the periphery of the nucleosome.

2. For the data in Figure 2A, does the distribution of viable spores in tetrads suggest that the decrease in spore viability is due to Meiosis I non-disjunction?

Yes, this is now included as a EV Figure 3B.

This would be expected if decreased Hop1 binding resulted in increased IS recombination and therefore fewer crossovers. While an increase in IS JMs was not observed at the *HIS4-LEU2* hotspot (although the JM data are not compelling), this single hotspot may not be indicative of the genome as a whole. If the spore viability pattern does not support MI non-disjunction as the reason for the dead spores, what ideas do the authors have for the basis of the spore inviability?

Increased IS recombination, as well as reduced DSBs can result in fewer COs and meiosis I non-disjunction. DSB mapping by Southern blot analyses (Figure 6,7, Appendix Figure S9) indicate lower DSBs among the *hop1-loop2* cells, and the loss of crossovers at *HIS4-LEU2* correlates directly with the loss of DSB activity at this locus. We have repeated the analysis of *HIS4-LEU2* and we do not detect any reduction in IH:IS bias compared to wild type. It is also worth noting that even a major loss in IH:IS bias (to well below 1, i.e. strongly favoring the sister) results in only a 50% loss of COs at the *HIS4-LEU2* locus (Lao et al., PLoS Genetics 2013). The fact that we observe a similar loss of COs in *hop1-loop2* mutants without apparent loss of IH:IS bias suggests that, at least at *HIS4-LEU2*, the drop in CO levels is not caused by defect in IH:IS bias. Nevertheless, it remains possible that at other loci IS:IH bias is affected. We have made this distinction explicit throughout the manuscript.

3. In Figure 3B, the authors assert that there is a "defect in chromosomal association of Hop1' since at the 2 hour timepoint the Hop1-loop2 protein covers only 20% of the nuclear area occupied by Hop1. However the values for both proteins at the 3 hour timepoint appear very similar and are perhaps elevated at 4 hours for Hop1-loop2. Therefore if there is a defect in chromosomal association it is overcome pretty rapidly. This conclusion should be weakened to reflect the data.

We have now weakened the conclusion here to reflect the data more accurately.

4. In Figure 4E put arrows on the insets indicating the spots corresponding to the IH and IS JMs. The signal for *hop1-loop2* is much fainter than WT and I don't see the IS JMs. The graph suggests that the JMs formed in *hop1-loop2* are similar to *hop1-loop2 pch2* but the JM spots for *hop1-loop2 pch2* are not convincing. How many times was this experiment done?

We have included additional 2D analyses to better represent the *hop1-loop2* phenotype (Figure 8B). We now note in the figure legend that the experiment was done three times.

Minor comments:

- Page 3: rather than "...Mek1, which enforces DSB repair via the homologous chromosome...", say "...Mek1, which biases DSB repair towards using the homologous chromosome..."
- , bottom: "...higher average rates of recombination..." should be higher average frequencies of recombination..." as a rate requires a time component (eg miles/hour)
- In the references, gene and genus species names should be italicized (eg. 38). Only the first word and proper nouns should be italicized (eg, 28).

The above 3 errors have been amended in the text.

Page 6, The text says that the ScHop1 CBR binds to nucleosomes with a Kd of 760 nM, but the table in Figure 1G say 0.5 uM. Shouldn't this be 0.76 uM?

Amended: the updated text reads "0.5 uM" as that is the final number.

Page 7, third paragraph, In wild-type cells...
In the fourth paragraph, wild-type Hop1..

In the updated manuscript we ensured to use "wild type" when type is the proper noun and "wild-type" when this term is used as an adjective.

Fourth paragraph, needs a citation for the statement that Zip1 is a transverse filament protein that is a marker for chromosome synapsis.

A citation has been added

The heading on page 8 should be *hop1-loop2* (italicized) spore survival and DSB formation depend (not depends) on *PCH2* (italicized). The experiment was genetic, not biochemical.

Amended

Page 9 first paragraph wild-type localization...

In the updated manuscript we ensured to use "wild type" when type is the proper noun and "wild-type" when this term is used as an adjective.

Figure 1 legend: For F, it should be indicated that the nucleosomes contain histones derived from humans

We have added this clarification to the legends of Figures 1C and 1F.

Figure 2A: Why are significance values not given for WT vs. loop2 tel and other combinations? Spore viability is highly quantitative and my guess from looking at the bar graphs is that the rescue of hop1-loop2 by tel1 is not complete (I agree with the authors that it is to "near WT levels"). The p-values should be shown to confirm this.

Amended to now include the significance values of all combinations.

It would be helpful if the green dots indicating centromeres in Figure 2C were more prominent.

The centromeres are now represented as triangles in all chromosome plots for better visualization.

In Figure 4C, why do some of the time points lack error bars? Were these timepoints only measured in one replicate? Similarly there are no error bars for the curves in 4D. How many times was this experiment repeated?

Amended to include error bars from the replicated experiments. The number of repeats are now indicated in the figure legend.

The authors found that the spore viability of hop1-loop2 is partially rescued by tel1, which they infer is because of an increase in DSB formation. It may be that the relatively high spore viability of the hop1-loop2 alone is also due to increased DSBs occurring due to the defect in synapsis, since synapsis is necessary for downregulating Spo11 activity (eg. Murakami, Keeney ecm11 paper).

We did consider the possibility that DSB formation is increased in the *hop1-loop2* mutants. To test this possibility, we now analyzed *zip1 pch2 hop1-loop2* triple mutants. DSB formation of the triple mutant was not elevated compared to the *pch2 hop1-loop2* double mutant (Figure 6, Appendix Figure S9), arguing that the further downregulation of DSB formation in the double mutant is not the result of increased synapsis. These data have been included in the revised manuscript.

Table S5: gene names should be italicized, the "a" in MATa should be in bold, H8644, LEU2/leu2Δ0. For the references, use the same numbering as the Bibliography. If blanks indicate strains that are unpublished, use "this work".
kanMX, not KanMX

H11569, remove LYS2/LYS2-genotypes for homozygous wild-type alleles are not included

Amended. (LEU2/leu2Δ0 is the appropriate nomenclature for S288c)

Figure S8. Have the p318-Hop1 antibodies been previously characterized and validated? If so, please supply a reference, if not, please show validation here.

Yes, this antibody was previously used in Subramanian et al., PLoS Biology 2016. We have added a reference for this antibody validation in the figure legend.

Why does the pHop1 antibody recognize a doublet in the hop1-loop2, pch2, and pch2 loop2 diploids? Only a single species is observed with the total Hop1 antibody.

Nice catch. We don't know why the pHop1 antibody is recognizing a doublet. We have amended the text to draw attention to this difference, but in the absence of additional clues, we chose not to speculate on the biological relevance of this in the text.

Referee #4:

The conserved meiotic axis protein protein Hop1 (and its homologs in other species) lies at the center of meiotic chromosome dynamics and recombination. As the penultimate component of the meiotic chromosome axis, Hop1 has critical roles in the initiation of meiotic recombination by double strand break formation, in the direction of meiotic DSB repair away from the sister chromatid and towards the homolog, and in modulating homolog pairing and synapsis. Most previous studies of Hop1 have focused on the role of Hop1's HORMA domain in recruiting Hop1 to the meiotic axis, via interaction with the meiotic axis protein Red1 (and analogs in other species), but Hop1 also has been shown to have DNA binding activity, suggesting that it is also recruited directly to chromatin. In the current work, Milano et al. combined structural biology, biochemistry, high-throughput ChIP-seq analysis, meiotic cytology and molecular analysis of meiotic recombination to identify roles for this second activity, residing in the C-terminal half of the protein, in Hop1 function. This is an important work that has the potential to substantially advance understanding in all organisms that undergo meiosis, and the data are compelling. I do have some disagreements regarding interpretation, but these most likely can be addressed by text revision, some addition analysis of the existing data, and possibly a few additional experiments that could readily be accomplished.

Major comments:

1. Hop1 binding to nucleosomes/DNA.

a. Previous work (predominantly by Muniyappa's group) has examined Hop1 DNA binding activity, but is not cited anywhere in the current manuscript. While Muniyappa's penultimate conclusion, that Hop1's physiological substrate is G-quadruplex, is most likely incorrect, there are still useful data in these papers and they ought to be at least briefly mentioned and discussed.

We apologize for this oversight. We have now amended the text (introduction) to include the work from Muniyappa's group showing direct DNA binding.

b. With regards to binding to nucleosomes vs. DNA, one notes that the Widom sequence used for structural and EMSA studies is quite G/C rich, and the question arises as to whether or not some kind of sequence content preferences are impacting where Hop1 is observed to bind, and the fact that under conditions of Hop1 excess the predominant species still only contains one Hop1 bound. While the Widom sequence is likely critical for getting a complex that is tractable for structural studies, it should not be necessary for EMSA studies. Authors do discuss this, but it would be useful to test the role of the specific target sequence in the measured affinities. Possible tests could include mutating the primary bound sequence to match the secondary bound sequence, shuffling sequences (while retaining G/C content), etc-up to the authors to decide. At the very least, Hop1 binding to a 40 nt oligo containing the secondary bound sequence needs to be done. It would also seem important, for the sake of completeness, to examine binding to DNA alone by the loop 2 mutant that abolishes nucleosome binding. It may turn out that previous studies of Hop1 binding to DNA will supply insight into appropriate substrates.

This may seem nit-picky and asking for a grab-bag of additional work, but given the argument that Hop1 preferentially binds to nucleosome-rich regions, it seems important to know whether the primary determinant of Hop1 binding to these regions is the density of nucleosomes per se, or underlying sequence characteristics that attract both Hop1 and nucleosomes independently. If this is not done, then the text presenting and discussing the data throughout the manuscript needs to be substantially revised to give more weight to possible sequence preference.

As we note above in a response to Reviewer #1, we chose not to pursue additional experiments with naked DNA substrates (either with Hop1 mutants or different DNA sequences) because overall, our data show that the most biologically relevant interactions are between the Hop1 CBR and nucleosome-wrapped, bent DNA. Therefore, we feel that there would not be significant biological insights gained from such experiments. We note that prior work from the Muniyappa lab has shown a preference for Hop1 binding to GC-rich versus AT-rich DNA (Kironmai et al., *MBoC* 1998). In contrast, our prior ChIP-Seq analysis showed that in cells, Hop1-bound genomic islands are more

nucleosome-rich than the genome as a whole but do not have elevated GC content (Heldrich et al., *NAR* 2022). This does of course not exclude other determinants in the DNA sequence, and we now added discussion of this possibility.

Regarding modifying the Widom sequence for EMSA studies (with bound histone proteins), we have attempted these experiments but found that we could not efficiently assemble nucleosomes with even minor modifications to the Widom sequence. Therefore, we could not confidently perform EMSA assays with mutated DNA sequences.

In addition, because in vitro binding experiments were done with the CBR rather than full-length Hop1, the question arises whether or not there are additional domains of Hop1 that contribute to chromatin binding. Addressing this may be pushing the bounds of the scope of the paper, but if it is not addressed then the appropriate caveats should be included.

Our data indicates that chromatin binding by Hop1 depends entirely on the CBR because no binding remains if the second recruitment pathway is removed (by deleting the cohesin *REC8*) (Figure 3B). That said, the reviewer is correct that we can't rule out effects of other domains in modulating chromatin binding. In cells, the HORMA domain is involved in binding the filamentous axis core protein Red1, which effectively results in oligomerization of Hop1 – this will no doubt affect chromatin binding affinity, but is difficult to address with in vitro experiments. We have added a discussion of these caveats to the end of the Results section titled “The Hop1 CBR binds nucleosomes”, and also further discuss possible effects of oligomerization in the discussion.

2. ChIP-Seq

a. From a cursory examination of the data in Figure 2C, it appears that the axis islands are not the only regions that are affected by loop 2 mutation. Because these are calibrated ChIPs, it should be possible to directly subtract the loop 2 mutant signal from wild type (similarly for other combinations), to identify regions that are differentially affected.

Thank you for this suggestion. Figure 2E now features such an analysis and includes a zoom in panel to emphasize regions that are differentially affected. We included whole genome plots of this analysis for *hop1-loop2* and *hop1-loop2 pch2* in Appendix Figure S7.

b. Figures 2D and 2E would be more readily interpreted if the Y-axis scales were the same in each panel-i.e. 1.0 to 3.5 in all panels in 2D, etc. It appears that the effect of loop 2 mutants is both to decrease axis island binding AND to increase non-island binding, in both PCH2 and pch2. If this is true, it would call for some re-interpretation. As a minor point, the colors used in these figure panels may not be color-blind friendly-something to check.

Figures are re-colored to ensure they are visible for color-blind readers. When appropriate, all Y-axis scales were made the same in all ChIP plots. Overall the Hop1-binding in *loop2* mutants is reduced (Figure 2B). Hop1 binding in *hop1-loop2* cells results in BOTH an increase in non-island binding, and a decrease in island binding, for *PCH2* and *pch2*. We have amended the text to reflect this.

3. Chromosome spreads versus ChIP-Seq.

The differences, in terms of impact of loop 2 and *pch2* mutation, between these two assays might be understood if ChIP-seq mostly measures the density of the Hop1 immediately adjacent to chromatin, while signal intensity on chromosome spreads, of course, detects every Hop1 monomer in the Hop1 chain. While this is considered briefly, it would be useful to discuss it more prominently, as many readers might not be aware of this important issue. For example, my impression is that the data might indicate that Pch2 is more active in remodeling/removing Hop1 in chains, and have less impact of chromatin-adjacent/bound Hop1.

In describing these results, we now note the difference between the chromatin association of Hop1 measured by ChIP and the localization of Hop1 assessed by IF.

Dr. Andreas Hochwagen
New York University
Department of Biology
New York 10003

2nd Jan 2024

Re: EMBOJ-2023-114485R
Chromatin binding by HORMAD proteins regulates meiotic recombination initiation

Dear Andreas and Kevin,

Thank you for submitting your revised manuscript to The EMBO Journal. It has now been seen once more by two of the original referees, and I am happy to say that both were generally satisfied with your revisions and responses to the initial comments. Following a final round of minor revision to incorporate referee 3's remaining suggestions, we should therefore be ready to accept the study for publication.

In addition, please make sure to fully address the following editorial issues during this final revision:

- On the abstract page of the manuscript, please include 4-5 general keyword terms to enhance searchability.
- Please also double-check all citations in the reference list, as several of them appear to be still incomplete (lacking page/locator numbers). Also, when citing preprints, please make sure to adhere to the format specified in our Guide to Authors: The citation in the text should be: "(preprint: NAME1 et al, YEAR)"; in the reference list: "Author NAME1, Author NAME2, ... (YEAR) article title. bioRxiv doi: XXX"
- Please rename the Conflict of Interest section into "Disclosure and Competing Interests Statement", in accordance with our updated Guide to Authors (<https://www.embopress.org/competing-interests/>)
- As we are switching from a free-text author contribution statement towards a more formal statement based on Contributor Role Taxonomy (CRediT) terms, please remove the present Author Contribution section and instead specify each author's contribution(s) directly in the Author Information page of our submission system during upload of the final manuscript. See <https://casrai.org/credit/> for more information.
- Please remove reviewer access information from the Data Availability section at this point, and ensure that data become publicly accessible upon acceptance.
- For the Expanded View figures, please adjust their callouts throughout the text from "EV Figure" to "Figure EV", and also make sure to always call out Appendix figures fully as "Appendix Figure S1/2/3..."
- For the 2 Expanded View tables, please adjust their callouts in the text as well as their names in the files to "Table EV1/2" (I note that Table EV1 is still named Table S4 inside the file). Also, please remove their legends from the main text file, and instead include them in the respective XLSX files, within a separate "Legend" tab.
- In the Appendix, please replace the word "Supplementary" with "Appendix" throughout the PDF, and make sure to head every table and every figure properly with "Appendix Table/Figure S1/2/3...". Ideally, please also place the legend for each Appendix Figure directly underneath the respective figure.
- Please provide suggestions for a short 'blurb' text prefacing and summing up the conceptual aspect of the study in two sentences (max. 250 characters), followed by 3-5 one-sentence 'bullet points' with brief factual statements of key results of the paper; they will form the basis of an editor-written 'Synopsis' accompanying the online version of the article. Please also upload a synopsis image, which can be used as a "visual title" for the synopsis section of your paper. The image (maybe based simply on Figure 10?) should be in PNG or JPG format, and please make sure that it remains in the modest dimensions of (exactly) 550 pixels wide and 300-600 pixels high.
- Please upload the Source Data files according to the scheme detailed in our previous communication: For each of the main figures, there should be one dedicated ZIP archive containing all respective files for this respective figure only. On the other hand, Source Data for all EV figures should be combined in one single ZIP archive, and uploaded as such. Similarly, Source Data for all Appendix figures should be combined in one single ZIP archive.
- Finally, during routine pre-acceptance checks, our data editors have raised the following queries regarding figures, data, and

legends:

- * Please indicate the statistical test used for data analysis in the legend of figure 2c.
- * Please define the box plots in terms of minima, maxima, centre, bounds of box and whiskers, and percentile in the legends of figures 5b-e.
- * Please add information related to n (number of replicates) in the legends of figures 5b-e.
- * Since n=2 biological replicates in Figure 6d, the error bars need to be removed, and both respective data points plotted individually (as meaningful calculation of variance is not possible for n<3)
- * Please define the error bars in the legends of figures 6b, d; 8a-b.
- * Figure legends for Figure 1F/Figure EV2 need to clearly state that the representative gel images in Fig. 1F are also shown in Fig. EV2

I am therefore returning the manuscript to you for a final round of minor revision, to allow you to make these modifications and upload the revised files. Once we will have received them, we should be ready to swiftly proceed with formal acceptance and production of the manuscript.

Yours sincerely,

Hartmut

- size of the scale bars that are mandatory for all micrograph panels
- the statistical test used to generate error bars and P-values
- the type error bars (e.g., S.E.M., S.D.)
- the number (n) and nature (biological or technical replicate) of independent experiments underlying each data point
- Figures may not include error bars for experiments with n<3; scatter plots showing individual data points should be used instead.

9) Digital image enhancement is acceptable practice, as long as it accurately represents the original data and conforms to community standards. If a figure has been subjected to significant electronic manipulation, this must be clearly noted in the figure legend and/or the 'Materials and Methods' section. The editors reserve the right to request original versions of figures and the original images that were used to assemble the figure. Finally, we generally encourage uploading of numerical as well as gel/blot image source data; for details see: embopress.org/page/journal/14602075/authorguide#sourcedata

At EMBO Press, we ask authors to provide source data for the main manuscript figures. Our source data coordinator will contact you to discuss which figure panels we would need source data for and will also provide you with helpful tips on how to upload and organize the files.

In the interest of ensuring the conceptual advance provided by the work, we recommend submitting a revision within 3 months (1st Apr 2024). Please discuss the revision progress ahead of this time with the editor if you require more time to complete the revisions. Use the link below to submit your revision:

Link Not Available

Referee #1:

The authors have addressed all of my concerns/comments appropriately. Hence, I am happy to recommend that the current manuscript is suitable for publication.

Referee #3:

The authors have adequately addressed all of my comments from the previous submission. I have only the following minor revisions that need to be made.

On page 7, the authors says a "2:2 (live:dead) spore segregation pattern is indicative of a Meiosis I defect. This is incorrect-that pattern would indicate the presence of a recessive lethal mutation. Meiosis I nondisjunction results in a change in the distribution of viable spores in tetrads from mostly 4:0, to a decreased number 4:0 tetrads, with an increase in 2:2 and 0:4 tetrads.

Page 8, 6 lines from the bottom: wild-type...mutants

Page 9, define FACS analysis

In Figure 8B, the purple line representing hop1-loop2 exhibits a lower ratio than the but WT line at all time points. It is difficult to distinguish between the error bars, but based on the averages, the authors are overstating the case when they say that the IS:IH ratio is comparable between WT and hop1-loop2 mutants.

Re: EMBOJ-2023-114485

“Chromatin binding by HORMAD proteins regulates meiotic recombination initiation”

Dear Referee,

Thank you again for your insightful and constructive feedback on our manuscript. We have taken special care to address your remaining points.

You will find our response to each raised concern in **Bold**.

Referee #3:

The authors have adequately addressed all of my comments from the previous submission. I have only the following minor revisions that need to be made.

On page 7, the authors says a "2:2 (live:dead) spore segregation pattern is indicative of a Meiosis I defect. This is incorrect-that pattern would indicate the presence of a recessive lethal mutation. Meiosis I nondisjunction results in a change in the distribution of viable spores in tetrads from mostly 4:0, to a decreased number 4:0 tetrads, with an increase in 2:2 and 0:4 tetrads.

We have now amended the text to be more accurate.

Page 8, 6 lines from the bottom: wild-type...mutants

We have amended this to ensure the appropriate use of wild-type vs. wild type

Page 9, define FACS analysis

We have now added text to define FACS

In Figure 8B, the purple line representing hop1-loop2 exhibits a lower ratio than the but WT line at all time points. It is difficult to distinguish between the error bars, but based on the averages, the authors are overstating the case when they say that the IS:IH ratio is comparable between WT and hop1-loop2 mutants.

The text is now modified to ensure our conclusion is not overstated.

Dr. Andreas Hochwagen
New York University
Department of Biology
New York 10003

11th Jan 2024

Re: EMBOJ-2023-114485R1
Chromatin binding by HORMAD proteins regulates meiotic recombination initiation

Dear Andreas,

Thank you for submitting your final revised manuscript for our consideration. I am pleased to inform you that we have now accepted it for publication in The EMBO Journal.

Yours sincerely,

Hartmut
